# JUDGELRM: LARGE REASONING MODELS AS A JUDGE

## ABSTRACT

Large Language Models (LLMs) are increasingly adopted as evaluators, offering a scalable alternative to human annotation. However, existing supervised fine-tuning (SFT) approaches often fall short in domains that demand complex reasoning. Judgment is inherently reasoning-intensive: beyond surface-level scoring, it requires verifying evidence, identifying errors, and justifying decisions. Through the analysis of evaluation tasks, we find a negative correlation between SFT performance gains and the proportion of reasoning-demanding samples, revealing the limits of SFT in such scenarios. To address this, we introduce **JudgeLRM**, a family of judgment-oriented LLMs, trained using reinforcement learning (RL) with judge-wise, outcome-driven rewards to activate reasoning capabilities. JudgeLRM consistently outperforms SFT-tuned baselines in the same size, as well as other RL and SFT variants, and even surpass state-of-the-art reasoning models: notably, on the human-generated out-of-distribution PandaLM benchmark, JudgeLRM-3B/4B surpasses its general-purpose teacher GPT-4, while JudgeLRM-7B/8B/14B outperforms DeepSeek-R1 by over 2% in F1 score, with particularly strong gains on reasoning-heavy tasks. Our findings underscore the value of RL in unlocking reasoning-aligned LLM judges. The code is available at https://anonymous.4open.science/r/JudgeLRM-D1C4/.

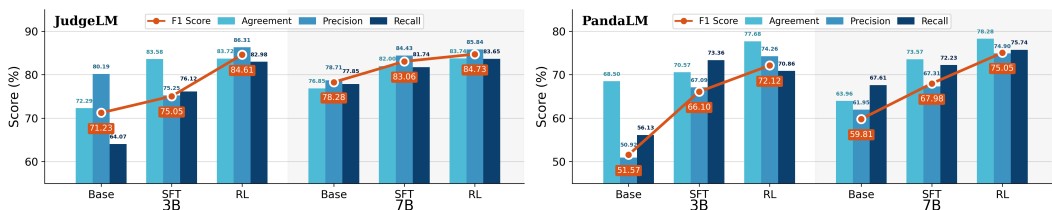

Figure 1: Performance improvement from base models (Qwen-2.5-3B/7B-Instruct) to SFT and to our judge-wise RL method on JudgeLM (left, in-distribution) and PandaLM (right, out-of-distribution). The fact that our RL method outperforms SFT suggests that judgment tasks are **reasoning-intensive**: they benefit more from *outcome-driven optimization* than *supervised imitation*.

## 1 INTRODUCTION

Recent advancements in LLMs have sparked significant interest in their use as evaluative judges (Gu et al., 2025; Li et al., 2024), offering a reliable and scalable alternative to costly human annotation. Previous approaches, such as JudgeLM (Zhu et al., 2025) and PandaLM (Wang et al., 2024), have enabled LLMs to perform judgment tasks based on the large scale SFT.

However, SFT faces inherent limitations in generalization and reasoning depth. **Do LLM judges truly require strong reasoning capabilities?** To explore this, we revisit prior work such as PandaLM (Wang et al., 2024), which shows that smaller models can be adapted for evaluation via post-training techniques. However, these methods still struggle with complex reasoning demands and computational inefficiency. We investigate whether improvements in judgment performance through SFT correlate with the proportion of questions that require reasoning to evaluate. Using the five question source categories defined in PandaLM (see Section A), we compute the proportion of samples in each category that require reasoning (detailed in Table 7). As shown in Figure 2, we observe an inverse relationship between improvements in F1 score on the judge task and the proportion of reasoning-required samples (labeled by GPT-4, elaborated in Appendix B), indicating that SFT

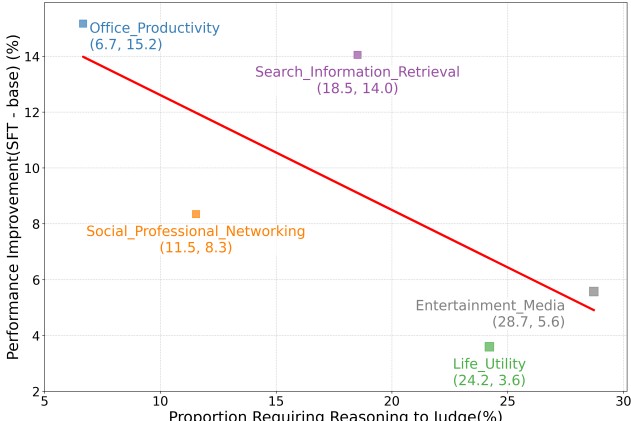

Figure 2: Judgment performance improvement vs. reasoning requirement across domainY-axis shows F1 score improvement (SFT – base) based on Qwen2.5-7B-instruct; X-axis shows the proportion of tasks requiring reasoning. Each point represents a domain. A negative linear trend (y = -0.41x + 16.72, R² = 0.53) suggests that domains with more reasoning-heavy tasks benefit less from SFT alone. Sample counts across domains: 105 (Office_Productivity), 108 (Search_Information_Retrieval), 195 (Entertainment_Media), and estimated 108 (Social_Professional_Networking), 190 (Life_Utility).

struggles more on categories demanding higher reasoning depth. Specifically, SFT gains drop as the proportion of reasoning-intensive items rises, implying that judges require flexible generalization skills rather than memorization abilities. This suggests that **effective LLM judges must possess strong reasoning abilities** to handle diverse and complex evaluation scenarios, where rote patterns from training memorization data fall short.

A fundamental challenge in deploying LLMs as judges lies in their dependence on intrinsic reasoning capabilities. While the Chain-of-Thought (CoT) (Wei et al., 2022) framework via SFT equips models to process nuanced information, achieving accurate and contextually grounded judgments remains non-trivial. Studies have shown that advanced large reasoning models (LRMs), such as DeepSeek-R1 (DeepSeek-AI, 2025), demonstrate strong performance in evaluation tasks by leveraging structured reasoning paths. To bridge this gap, we propose **JudgeLRM**, a family of LLMs trained using RL with judge-wise, outcome-driven rewards to enhance evaluative reasoning. The task-specific reward function integrates both structural and content-based components. The structural reward ensures well-formatted reasoning and answer sections, while the content reward aligns model judgments with ground-truth preferences through relation, absolute, and confidence-based metrics. This design promotes both faithful reasoning and accurate, confident scoring.

JudgeLRM model series (licensed under MG0-2.0[1]) ranges from 3B to 14B parameters, trained using Group Relative Policy Optimization (GRPO) (Shao et al., 2024). Empirical results demonstrate that JudgeLRM not only surpasses proprietary models like GPT-4 and DeepSeek-R1 (Table 2) but also outperforms SFT and RL baselines of comparable sizes (Figure 1), with an average improvement of 8.14% in F1 score over SFT counterparts. Notably, JudgeLRM shows consistent gains even in categories with a high proportion of reasoning-required tasks (Figure 4), further illustrating its ability to overcome SFT's reasoning limitations. These implications highlight that **judgment is inherently a reasoning-intensive task**, not merely a scoring exercise, and that RL-based approaches can effectively instill the flexible generalization needed for robust judge models. Through systematic analysis, we demonstrate that reasoning patterns, such as verification, sub-goal setting, double checking, error identification, and decision justification, are crucial to the success of judgment tasks.

## 2 RELATED WORK

### 2.1 LLM REASONING AND PLANNING

Modern large language models (LLMs) demonstrate remarkable reasoning abilities through their intrinsic Chain-of-Thought (CoT) mechanisms (Wei et al., 2022). This capability can be elicited

---

[1]https://www.modelgo.li/

through prompting techniques (Kojima et al., 2022) or explicitly integrated into model architectures like Openai o1 (OpenAI, 2024), Deepseek R1 (DeepSeek-AI, 2025; Shao et al., 2024). Such structural enhancements enable LLMs to perform multi-step reasoning across diverse domains, like in medical, Huatuo-o1 (Chen et al., 2024b), Medical-R1 (Lai et al., 2025) and in finance, Fin-o1 (Qian et al., 2025) and Fin-R1 (Liu et al., 2025a). While models like DeepSeek-R1 demonstrate strong reasoning in mathematics and coding, these domains typically involve **deductive reasoning** within closed-ended problems with definitive, verifiable solutions. (Huang & Chang, 2023) In contrast, the task of evaluating open-ended text involves **abductive reasoning** (Peirce, 1955). Judging requires the model to identify subtle quality nuances and generate its explanation (e.g. identification, justification) (Walton, 2014), rather than the rigid logical chains of mathematical derivation. Our work **first focuses on energizing reasoning abilities for judges**, subsequently revealing the inherently reasoning-intensive nature of the judging task and validates the necessity and effectiveness of strong reasoning skills for judges.

## 2.2 LLM AS A JUDGE

Human evaluation of LLM outputs is time-consuming, resource-intensive, and often inconsistent due to annotator subjectivity (Gu & Others, 2024). To address these limitations, researchers have explored using LLMs themselves as evaluators, a paradigm often referred to as 'LLM-as-a-Judge' (Zheng et al., 2024). This approach promises more scalable and potentially cost-effective evaluation. Furthermore, recognizing the potential benefits of specialized models, some studies have focused on training dedicated LLMs specifically for the task of judging LLM outputs, aiming for improved accuracy and alignment with human preferences (Zhu et al., 2025; Wang et al., 2024). Despite its promise, the LLM-as-a-Judge approach faces various biases inherent in the judge LLMs themselves, which can compromise the fairness and reliability of the evaluations (Gallegos et al., 2024; Chen et al., 2024a; Dubois et al., 2025).

To overcome SFT limitations, advanced preference modeling methods have been explored, utilizing RLHF (Ouyang et al., 2022) for general alignment and NLHF (Munos et al., 2024) for signal aggregation. However, these approaches typically rely on discriminative Bradley-Terry models that produce immediate, intuitive scalar predictions ("System 1"), limiting their capacity for the multi-step verification required in complex evaluations. By leveraging "System 2" paradigms to integrate explicit reasoning (Wei et al., 2022), we **first introduce RL with GRPO (DeepSeek-AI, 2025) for training judge models**. While concurrent works such as J1 (Whitehouse et al., 2025), RM-R1 (Chen et al., 2025) also utilize RL to incentivize thinking, JudgeLRM occupies a distinct niche by focusing on **efficient emergence** in compact models. We demonstrate that specialized *outcome-driven rewards* alone are sufficient to activate robust reasoning behaviors in small-scale models as a judge without process supervision, demonstrating results comparable to Deepseek-R1 and a controllable range of bias.

## 3 JUDGE-WISE OUTCOME REWARD AND RL TRAINING

Inspired by the insufficiency of SFT training revealed in Fig. 2, we introduce Large Reasoning Models as a judge (JudgeLRM), applying RL on judge tasks through judge-wise outcome reward.

### 3.1 JUDGE TASK DEFINITION

We consider a judge task where, given a query $Q$ and two responses $A_1$ and $A_2$, the model evaluates them on a scale of 1–10 $(s_1, s_2)$, following the JudgeLM (Zhu et al., 2025) setting. Evaluation criteria include helpfulness, relevance, accuracy, and level of detail, as specified in Prompt 3. The goal is to predict scores that align with human judgments, which determine whether $A_1$ wins, ties with, or loses to $A_2$.

### 3.2 REWARD DESIGN FOR JUDGE TASK

Rule-based rewards have shown strong performance and are widely adopted (DeepSeek-AI, 2025). For judge tasks involving reasoning, we design a reward function that combines structural and

---

**System Prompt for RL Training**

```
<|im_start|>system
```
You are a helpful assistant. The assistant first performs a detailed, step-by-step reasoning process in its mind and then provides the user with the answer. The reasoning process and answer are enclosed within `<think> </think>` and `<answer> </answer>` tags, respectively, i.e., `<think>` detailed reasoning process here, explaining each step of your evaluation for both assistants `</think><answer>` answer here `</answer>`. Now the user asks you to judge the performance of two AI assistants in response to the question. Score assistants 1-10 (higher=better). Criteria includes helpfulness, relevance, accuracy, and level of detail. Avoid order, length, style or other bias. After thinking, when you finally reach a conclusion, clearly provide your evaluation scores within `<answer> </answer>` tags, i.e., for example,`<answer>3</answer><answer>5</answer>`
```
<|im_end|>
<|im_start|>user
[Question]
{question}

[Assistant 1's Answer]
{answer_1}

[Assistant 2's Answer]
{answer_2}
<|im_end|>
<|im_start|>assistant
<think>
```

Figure 3: System Prompt for RL Training

content-based components. Specifically, the reward $r$ for sample $i$ is defined as $r_i = \mathcal{R}_{\text{struct}} + \mathcal{R}_{\text{content}}$, inspired by Xie et al. (2025).

**Structural Reward** As shown in Figure 3, the structural reward $\mathcal{R}_{\text{struct}}$ ensures that the model output includes a structured reasoning process enclosed in `<think>...</think>` tags and final judgment scores $(s_1, s_2)$ enclosed in `<answer>...</answer>` tags, where $s_1, s_2 \in \{1, \ldots, 10\}$. The reward components are:

$$\mathcal{R}_{\text{format}} = \begin{cases} 1.0, & \text{if all tags are correct and in proper order} \\ -0.5, & \text{if } s_1, s_2 \notin \{1, \ldots, 10\} \\ -1.0, & \text{if severe formatting errors exist} \end{cases} \quad (1)$$

**Content Reward** The content reward $\mathcal{R}_{\text{content}}$ evaluates the accuracy and confidence of the predicted scores $(s_1, s_2)$ against the ground-truth labels $(s_1^*, s_2^*)$. It consists of three parts:

*Relation Reward* in Eq. 2 ensures the model correctly ranks responses, aligning with the core goal of **comparative evaluation**. Aligning relative order is more important than absolute score matching in evaluation tasks, because win-loss relationships are more concerned, so the value is dominant in the content reward.

$$r_{\text{relation}} = \begin{cases} 2.0, & \text{if } \text{sgn}(s_1 - s_2) = \text{sgn}(s_1^* - s_2^*) \\ -1.5, & \text{otherwise} \end{cases} \quad (2)$$

*Absolute Reward* in Eq. 3 refines score-level accuracy, tolerating minor errors while penalizing large deviations. The scores are gradually refined by hierarchical rewards (complete match $\rightarrow$ partial match $\rightarrow$ no reward) to solve '**sparse rewards**' in reinforcement learning: if relying only on relational rewards, the model may learn only coarse sequential judgements and ignore score calibration.

$$r_{\text{absolute}} = \begin{cases} 1.0, & \text{if } |s_1 - s_1^*| + |s_2 - s_2^*| = 0 \\ 0.6, & \text{if } r_{\text{relation}} = 2 \text{ and } |s_1 - s_1^*| + |s_2 - s_2^*| \leq 2 \\ 0, & \text{otherwise} \end{cases} \quad (3)$$

*Confidence Reward* in Eq. 4 promotes decisive judgments when the ranking is correct, avoiding ambiguous scores. **Harder judge problems result in ambiguous scores**, so we stimulate high confidence score to **improve 'decision confidence'**, which results in more polarised decision output. We activate it only when the relative order is correct to avoid the model blindly expanding the variance.

To avoid over-polarization from continous rewards (in Table 3), the discrete confidence reward acts as a reverse hinge loss, saturating at a target level to curb extreme judgments (see Appendix F.1).

$$r_{\text{confidence}} = \begin{cases} 0.2, & \text{if } r_{\text{relation}} = 2 \text{ and } |s_1 - s_2| \geq |s_1^* - s_2^*| \\ 0, & \text{otherwise} \end{cases} \tag{4}$$

The final reward $r(s_1, s_2)$ is computed as:

$$r(s_1, s_2) = \underbrace{\mathcal{R}_{\text{format}}}_{\mathcal{R}_{\text{struct}}} + \underbrace{r_{\text{relation}} + r_{\text{absolute}} + r_{\text{confidence}}}_{\mathcal{R}_{\text{content}}} \tag{5}$$

The design of structural and content rewards encourages the model to learn **distinguishing** good/bad and make relative judgments, **rather than pursuing absolute high rewards**, which is neglected by previous SFT methods on judges (Wang et al., 2024; Zhu et al., 2025). We discuss the sensitiveness to reward exact values and the relative reward relationship in RL rule-based training in Appendix F.

### 3.3 RL TRAINING ALGORITHM

We utilize GRPO (DeepSeek-AI, 2025) as the policy gradient algorithm, which eliminates the need for a separate value model in Proximal Policy Optimization (PPO) (Schulman et al., 2017) by grouping samples and normalizing **intra-group** advantages for more practical training. Moreover, it reduce the data collection cost and increases generalizability than SFT because it does not need explicit reasoning chain to learn. For policy $\pi_\theta$ and reference model $\pi_{\text{ref}}$, we define:

**Group-wise Advantage Calculation**  For each query group $G_Q$ with its associated judgments $G_Q = \{(A_1, A_2, s_1, s_2)\}$, the mean $\mu_Q$ and the standard deviation $\sigma_Q$ of reward $r$ in group $Q$ is calculated as:

$$\mu_Q = \mathbb{E}_{(A_1, A_2) \sim G_Q}[r(s_1, s_2)], \quad \sigma_Q = \sqrt{\mathbb{E}_{(A_1, A_2) \sim G_Q}[(r(s_1, s_2) - \mu_Q)^2]} \tag{6}$$

The normalized advantage $\mathcal{A}$ for sample $i$ is defined for the optimization below to quantify how good $i$ is relative to the average action:

$$\mathcal{A}_i(s_1, s_2|Q) = \frac{r_i(s_1, s_2) - \mu_Q}{\sigma_Q + \eta} \tag{7}$$

In which $\eta$ is a smoothing term for numerical stability during training, preventing $\sigma_Q$ from being too small during normalization (Schulman et al., 2017). By normalizing intra-group advantages $\mathcal{A}$, variance is reduced to improve the **stability** of policy updates. This allows the model to receive an effective learning signal even on difficult tasks where rewards are generally low, thereby **alleviating the training imbalance** caused by varying task difficulties across domains in Figure 2.

**Policy Optimization Objective**  The policy $\pi_\theta$ of parameter $\theta$ is optimized using the following objective:

$$J_{\text{GRPO}}(\theta) = \mathbb{E}_{Q \sim \mathcal{D}} \mathbb{E}_{(s_1, s_2) \sim \pi_{\text{old}}} \left[ \min \left( \frac{\pi_\theta(s_1, s_2|Q, A_1, A_2)}{\pi_{\text{old}}(s_1, s_2|Q, A_1, A_2)} \mathcal{A}_i(s_1, s_2|Q), \right. \right.$$
$$\left. \left. \text{clip}\left( \frac{\pi_\theta(s_1, s_2|Q, A_1, A_2)}{\pi_{\text{old}}(s_1, s_2|Q, A_1, A_2)}, 1 - \epsilon, 1 + \epsilon \right) \mathcal{A}_i(s_1, s_2|Q) \right) - \beta D_{\text{KL}}(\pi_\theta \| \pi_{\text{ref}}) \right] \tag{8}$$

In which the clip range $\epsilon$ limits the magnitude of policy updates to prevent training instability, and the KL divergence is set to prevent $\pi_\theta$ from excessively deviating from reference model $\pi_{\text{ref}}$ with factor $\beta$ (Schulman et al., 2017).

## 4 EXPERIMENTS

We empirically evaluate JudgeLRM to address the following research questions:

- **Effectiveness** : How does JudgeLRM perform compared to other SFT and RL baselines (Section 4.2) and state-of-the-art LRMs and specialized judgement models (Section 4.3)?
- **Ablation Study** (Section 4.4): What are the individual contributions of different reward components to JudgeLRM's performance?
- **Reliability** (Section 4.5): Can JudgeLRM provide consistent and accurate comparative judgments?
- **Case Study** (Section 4.7): In what ways does JudgeLRM demonstrate effective reasoning to support high-quality judgments?

## 4.1 EXPERIMENTAL SETUP

Our experiments mainly focus on pair-wise judge scenarios (since we also support other scenario as discussed in Section 4.3), targeting on open-ended QA evaluation rather than closed-domain or expert-level tasks, and JudgeLM (Zhu et al., 2025) PandaLM (Wang et al., 2024), aligning directly with our objective. By contrast, benchmarks such as RewardBench (Lambert et al., 2024), JudgeBench (Tan et al., 2025), while valuable, primarily address single-judge or domain-specific evaluation scenarios that fall outside the scope of this study. Our design emphasizes isolating the contribution of reasoning, rather than data scaling or architectural engineering, by assessing how RL enhances the generalization of judges over SFT, under the same dataset and base-model conditions as JudgeLM. Following prior work (Zheng et al., 2024), we prioritize controlled experiments on well-established benchmarks to ensure comparability.

**Datasets.** We evaluate on two established benchmarks. JudgeLM uses GPT-4 annotations as gold labels, comprising 100K training instances and a 5K test set, following the task formulation described in Section 3.1. PandaLM provides human-annotated pairwise comparisons with win/tie/loss labels on a 1K test set, offering complementary supervision. Together, these datasets allow us to assess performance across both GPT-aligned and human-grounded evaluation settings.

**Baselines.** We compare JudgeLRM against three categories of baselines: (1) *Base, SFT and RL methods*: Base (Table 1), SFT (Table 1,1), Bradley-Terry (Bradley & Terry, 1952), Direct Preference Optimization (DPO) (Rafailov et al., 2023), etc (Table 1). (2) State-of-the-art proprietary LLMs, including GPT-3.5, GPT-4, and Deepseek-R1 (Table 2); (3) *Specialized judgment models*: Auto-J-13B, JudgeLM-7B/13B/33B, and PandaLM-70B (Table 2). This setup enables a fair comparison across different alternatives, isolating the effect of reasoning-oriented training.

**Evaluation Metrics.** Following prior work, we evaluate model performance using agreement-based metrics: precision, recall, and F1 score,which quantify alignment with teacher model judgments.

**Implementation Details.** Our models, JudgeLRM-3B/7B/14B and JudgeLRM-4B/8B, are based on Qwen2.5-Instruct and Qwen3, respectively. All models including baselines are trained exclusively using reinforcement learning on the JudgeLM training set. The training is conducted over a single epoch, with a maximum prompt length of 1024 tokens and a maximum response length of 2048 tokens, using a batch size of 16. JudgeLRM-3B/4B is trained on 4×A100 80GB GPUs with a learning rate of 3e-7, while JudgeLRM-7B/8B/14B is trained on 8×A100 80GB GPUs with a learning rate of 1e-6. Following Xie et al. (2025)[2], the group size of $Q$ is set to 8, the clip range $\epsilon$ is set to 0.5, the KL divergence factor $\beta$ is set to 0.001, and smooth term $\eta$ is set to 1e-6.

## 4.2 BASELINES: ADDITIONAL SFT AND RL VARIANTS

| Dataset | PandaLM (Human as ground truth. Out of distribution.) | | | |
|---|---|---|---|---|
| **Criteria** | **Agreement** | **Precision** | **Recall** | **F1** |
| *Method Baselines. (trained JudgeLM)* | | | | |
| Qwen2.5-3B-Instruct | 68.50 | 50.92 | 56.13 | 51.57 |
| Qwen2.5-3B-Instruct-Judge-SFT | 70.57 | 67.09 | 73.36 | 66.10 |
| DPO-Answer (Qwen2.5-3B-Instruct) | 67.27 | 44.79 | 50.07 | 47.27 |
| CLS-RM (Qwen2.5-3B) | 58.15 | 51.70 | 51.69 | 51.69 |
| Bradley-Terry (Qwen2.5-3B) | 58.94 | 59.04 | 59.06 | 58.94 |
| Cross-BT (Qwen2.5-3B) | 75.18 | 50.06 | 55.97 | 52.84 |
| SFT-Think(Qwen2.5-3B-Instruct) | 72.49 | 81.69 | 56.01 | 55.03 |
| SFT-Distill-R1-Think (Qwen2.5-3B-Instruct) | 73.37 | 66.45 | 70.28 | 67.33 |
| DPO-RC (Qwen2.5-3B-Instruct) | 68.67 | 64.54 | 69.43 | 65.23 |
| *Ours.* | | | | |
| JudgeLRM-3B | 77.68 | 74.26 | 70.86 | 72.12 |

Table 1: Comparison with other RL and SFT methods.

To provide a comprehensive comparison, we construct several additional baselines using the Qwen2.5-3B family (base model in parentheses). These methods reflect alternative SFT or RL training paradigms but differ from our JudgeLRM design in how they handle preferences and reasoning. Appendix D provides Formalization of Baseline and JudgeLRM Objectives.

**(1) DPO-Answer-3B (Qwen2.5-3B-Instruct).** Optimizes a generative policy via DPO on chosen/rejected pairs using only win/loss signals, predicting preferences by comparing response likelihoods, focusing on generation preference rather than explicit judgment.

---

[2]Realization (Xie et al., 2025) of Equation 8 (DeepSeek-AI, 2025) does not include response-level length bias (Liu et al., 2025b) in origin GRPO (Shao et al., 2024).

| Dataset | JudgeLM (GPT-4 as ground truth. In distribution.) | | | | PandaLM (Human as ground truth. Out of distribution.) | | | |
|---|---|---|---|---|---|---|---|---|
| Criteria | Agreement | Precision | Recall | F1 | Agreement | Precision | Recall | F1 |
| *Baselines. (* from original paper)* | | | | | | | | |
| GPT-3.5* | 73.83 | 70.70 | 52.80 | 52.85 | 62.96 | 61.95 | 63.59 | 58.20 |
| GPT-4* | - | - | - | - | 66.47 | 66.20 | 68.15 | 61.80 |
| PandaLM-7B | 68.61 | 40.75 | 38.82 | 39.41 | 59.26 | 57.28 | 59.23 | 54.56 |
| Auto-J-13B* | 74.86 | 61.65 | 57.53 | 58.14 | - | - | - | - |
| JudgeLM-7B | 81.11 | 69.67 | 78.39 | 72.21 | 65.07 | 66.89 | 71.95 | 61.92 |
| JudgeLM-13B* | 84.33 | 73.69 | 80.51 | 76.17 | 68.97 | 68.21 | 74.15 | 65.12 |
| JudgeLM-33B* | **89.03** | 80.97 | 84.76 | 82.64 | 75.18 | 69.30 | 74.93 | 69.73 |
| PandaLM-70B* | - | - | - | - | 66.87 | 74.02 | 66.87 | 69.23 |
| Deepseek-R1 | - | - | - | - | 78.67 | 77.51 | 69.97 | 72.48 |
| *Ours. (Qwen3 Base)* | | | | | | | | |
| JudgeLRM-4B | 84.61 | 86.82 | 84.24 | 85.51 | 77.88 | 72.33 | 73.67 | 72.87 |
| JudgeLRM-8B | 86.69 | 89.16 | 85.76 | 87.43 | 80.78 | 78.25 | 73.78 | 75.54 |
| JudgeLRM-4B-single | 81.17 | 80.13 | 86.61 | 83.24 | 64.56 | 58.60 | 61.70 | 58.50 |
| JudgeLRM-8B-single | 83.20 | 81.64 | **88.84** | 85.09 | 71.27 | 65.07 | 69.34 | 65.30 |
| *Ours. (Qwen2.5-Instruct Base)* | | | | | | | | |
| JudgeLRM-3B | 83.72 | **86.31** | 82.98 | 84.61 | 77.68 | 74.26 | 70.86 | 72.12 |
| JudgeLRM-7B | 83.74 | 85.84 | 83.65 | 84.73 | 78.28 | 74.90 | **75.74** | 75.05 |
| JudgeLRM-14B | 85.25 | 86.64 | 85.91 | 86.27 | **81.18** | **78.39** | 74.80 | **76.29** |

Table 2: Performance on JudgeLM and PandaLM. As pairwise comparisons rarely yield ties, we exclude tie cases (∼10% of the test set) for more interpretable evaluation, which JudgeLRM-7B's F1 reaches 83.47, all metrics on PandaLM surpass DeepSeek-R1 (see Table 11 for details). More analysis of results and performance visualization are presented in Appendix H.

**(2) CLS-RM-3B (Qwen2.5-3B).** Fine-tunes a reward model (RM) with a classification head. The RM assigns absolute scores to individual responses, and preferences are inferred by comparing these scores. Unlike JudgeLRM, this approach does not leverage relational reasoning across responses.

**(3) Bradley-Terry-3B (Qwen2.5-3B).** Implements a pairwise preference model following the Bradley–Terry (BT) formulation, trained on chosen/rejected pairs with associated preference scores. The model predicts a scalar reward for each response, which is used for pairwise evaluation.

**(4) Cross-BT-3B (Qwen2.5-3B).** Implements a single-input pairwise BT Model. A cross-encoder takes the full triplet $(Q, A_1, A_2)$ as a single input and optimizes a binary cross-entropy loss using soft labels derived from the score difference, aligning with the standard BT formulation.

**(5) SFT-Think-3B (Qwen2.5-3B-Instruct).** SFT using a JudgeLRM-style template where the model generates an explanation enclosed in `<think>` tags before producing its judgment. This setting isolates SFT under structural constraints. We find explanations fail to function as effective reasoning chains, leading to weaker adherence to structure and underperformance compared to SFT baselines.

**(6) SFT-Distill-R1-Think (Qwen2.5-3B-Instruct).** SFT using high-quality reasoning traces distilled from DeepSeek-R1 to investigate the limits of data distillation.

**(7) DPO-RC-3B (Qwen2.5-3B-Instruct).** Additional RL baseline following SPIN (Chen et al., 2024c; Verl Community, 2025), trained with a content-based reward (RC, $\mathcal{R}_{\text{content}}$) while adopting the JudgeLRM-style data template. This method performs competitively with SFT-trained models but still lags behind JudgeLRM-3B.

Overall, (1-3) remain limited because they abstract judgment into *token-level* preference modeling, losing *relational understanding* necessary for comparative evaluation and the ability to provide *rationale*. While (4) achieves relatively high agreement, its low Precision and F1 indicate that it overfits to surface-level heuristics instead of judging content quality. (5) highlights SFT's structural weakness, *formatting* instructions alone do not yield robust reasoning chains. (6) indicates that imitation of reasoning traces alone fails to capture the intrinsic dynamics. (7) lacks the structural reward ($\mathcal{R}_{\text{struct}}$), which is crucial for enforcing the explicit reasoning process (like verification and double-checking) required for high-fidelity evaluation. Furthermore, all (1-7) do not benefit from the group advantage's stability of JudgeLRM across heterogeneous task difficulties.

### 4.3 JUDGE PERFORMANCE

We benchmark JudgeLRM against a range of competitive models, with results summarized in Table 2. At the 7B scale, JudgeLRM-7B substantially outperforms its SFT counterpart (Qwen2.5-7B-Instruct-Judge-SFT) as well as the task-specific JudgeLM-7B model. Remarkably, **JudgeLRM-**

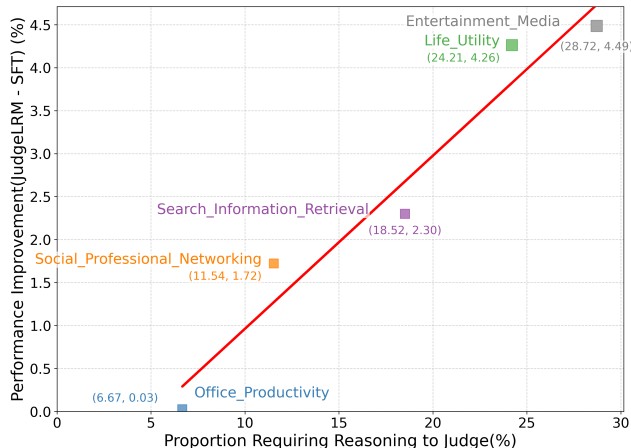

Figure 4: Judgment performance improvement vs. reasoning requirement across domains. The Y-axis indicates the F1 score improvement of JudgeLRM-7B over the Qwen2.5-7B-Instruct-Judge-SFT baseline; the X-axis represents the proportion of tasks within each domain that require reasoning. Each point corresponds to a domain. A positive linear trend ($y = 0.2x - 1.05$, $R^2 = 0.95$) suggests that domains with a higher proportion of reasoning-intensive judge tasks see greater performance gains from JudgeLRM-7B. See Appendix G for further analysis.

**3B/4B surpasses GPT-4 on the human-annotated PandaLM benchmark, and scaling further to JudgeLRM-7B/8B matches or even surpass the performance of DeepSeek-R1.** Moreover, we also showcase JudgeLRM's ability to judge single answer in "-single" setting in Table 2 with first testing score through prompt 5 and pairwise judge test like CLS-RM. It surpasses all baselines with the same size, showing the adaptability of JugdeLRM.

To investigate the source of these gains, we examine JudgeLRM-7B's behavior on reasoning-intensive subsets of PandaLM. Figure 4 shows a positive correlation between performance improvements and the proportion of reasoning-demanding instances. Specifically, when comparing JudgeLRM-7B against its base model (Qwen2.5-7B-Instruct), we observe a correlation coefficient of 0.20 between relative improvement and reasoning rate, with larger gains concentrated in reasoning-heavy categories. This trend provides direct evidence that judgment is inherently a reasoning-intensive task, and that reinforcement learning with outcome-driven rewards equips JudgeLRM with stronger evaluative reasoning than SFT-based baselines.

## 4.4 ABLATION STUDY

To disentangle the effect of different reward components, we perform an ablation study on the human-annotated PandaLM benchmark. Our analysis focuses on the content-based rewards introduced in Section 3.2, namely $r_{absolute}$ and $r_{confidence}$. These rewards explicitly encourage accurate scoring and calibrated confidence, thereby guiding models to verify their reasoning chains and revise errors within the <think> step before finalizing a judgment.

As shown in Table 3, removing content rewards leads to a consistent 2–5% drop in F1 score. Without these signals, models tend to produce superficial explanations or fail to detect inconsistencies between evidence and verdict, underscoring that judgment accuracy requires more than structural formatting alone. It demonstrates that outcome-driven content rewards are crucial for eliciting faithful reasoning, effective error correction, and reliable decision-making in JudgeLRM.

**Effect of Incentivizing Lengthy response** During JudgeLRM training, we observe that both reasoning and response lengths naturally increase with training steps, with larger models (e.g., 7B) producing longer outputs than smaller ones (3B) (Appendix K, Figure 22). To test whether explicitly encouraging longer reasoning chains improves performance, we introduced the following length reward:

| Test on PandaLM | Agreement | Precision | Recall | F1 |
|---|---|---|---|---|
| JudgeLRM-7B | **78.28** | 74.90 | **75.74** | **75.05** |
| - w/o. $r_{rel}$ | 52.55 | 49.95 | 50.61 | 46.43 |
| - w. $r_{rel}$, w/o. $r_{abs} + r_{conf}$ | 75.78 | 69.09 | 73.69 | 70.36 |
| - w. $r_{rel} + r_{conf}$, w/o. $r_{abs}$ | 75.58 | 70.64 | 66.69 | 68.16 |
| - w. $r_{rel} + r_{abs}$, w/o. $r_{conf}$ | 77.08 | 71.36 | 75.14 | 72.36 |
| - w. continuous $r_{conf} = 0.1 \times \Delta$) | 74.37 | 62.52 | 57.19 | 56.12 |
| - w. $\mathcal{R}_{length}$ ($L = 120$) | **78.28** | **75.81** | 69.19 | 71.34 |
| - w. $\mathcal{R}_{length}$ ($L = 300$) | 76.98 | 73.80 | 70.68 | 71.98 |

Table 3: Ablation study of 7B models on the human-annotated PandaLM dataset.

| Methods | Consistency ↑ (w/ swap.) | Bias ↓ toward 1st | Bias ↓ toward 2nd | Delta Bias ↓ |
|---|---|---|---|---|
| JudgeLM score-first* | 73.45 | 19.83 | 6.72 | 13.11 |
| GPT-4* | 85.82 | 5.55 | 3.09 | 2.46 |
| Qwen-7B-Instruct 0shot | 77.11 | 12.96 | 9.93 | 3.04 |
| JudgeLRM-7B | 84.50 | 5.39 | 10.11 | 4.72 |
| JudgeLRM-8B | 89.55 | 4.04 | 6.40 | 2.36 |

Table 4: Assessment of position bias on the *val* split of JudgeLM. We evaluate model self-consistency under different answer order permutations, quantify bias toward the first or second answer, and report the gap between these biases ($\Delta_{bias}$).

$$\mathcal{R}_{length} = \begin{cases} 0.2, & \text{if the reasoning chain exceeds } \mathbf{L} \text{ tokens} \\ -1.0, & \text{if the maximum token limit is reached} \\ 0, & \text{otherwise} \end{cases} \tag{9}$$

However, as shown in Table 3, simply incentivizing slightly longer answers degraded results of about 3%, suggesting that merely increasing output length (potentially via repetition) **does not benefit** the judge task.

## 4.5 RELIABILITY OF MODEL JUDGMENT

A key property of judgment models is reliability – the ability to produce consistent and accurate judgments regardless of the order in which candidate answers are presented. To evaluate the reliability of JudgeLRM and representative baselines, we conduct experiments on the JudgeLM dataset by permuting the order of answers. We assess three aspects: (1) self-consistency across permutations, (2) positional bias toward the first or second answer, and (3) the bias gap ($\Delta_{bias}$), which captures variance in position-based preferences.

As shown in Table 4, JudgeLRM substantially improves self-consistency compared to both its base Instruct model and the task-specific JudgeLM baseline. Additionally, JudgeLRM significantly reduces bias toward the first answer while maintaining comparable bias levels toward the second, achieving more balanced and robust evaluation. It demonstrate that reinforcement learning with outcome-driven rewards not only enhances reasoning ability but also mitigates order sensitivity.

## 4.6 REASONING PATTERN ANALYSIS

To further understand how JudgeLRM achieves superior performance, we analyze the capabilities in reasoning steps, specifically Deduction, Induction, and Abduction (detailed in Appendix B), often referred to as *meta-abilities* (Gandhi et al., 2025; Hu et al., 2025), extracted using a strengthened prompt with gpt-5-nano-2025-08-07 to avoid over-interpreting descriptive sentences.

As shown in Table 5, JudgeLRM-3B outperforms the SFT-Distill model significantly in **Abduction** (+8.5%). While SFT successfully mimics the "linear narrative" style of the teacher model (high Induction), it lacks depth in justification. The results suggest that the outcome-driven RL reward forces the model to engage in genuine pattern recognition and trial-and-error to maximize judgment accuracy, rather than merely imitating the surface form of reasoning. This trend holds at the 8B scale, where JudgeLRM-8B shows a massive jump in Deduction (+17.5%), validating that RL effectively activates the capability to verify criteria against prompt strictness.

| Model (%) | Deduction (e.g., Verification) | Induction (e.g., Synthesis) | Abduction (e.g., Justification) |
|---|---|---|---|
| Qwen2.5-3B-Instruct | 12.4 | 48.8 | 15.8 |
| SFT-Distill-R1-Think-3B | 31.2 | 61.6 | 26.6 |
| JudgeLRM-3B | 33.2 | 57.2 | **35.1** |
| Qwen3-8B | 44.4 | 58.9 | 39.1 |
| JudgeLRM-8B | **61.9** | 61.7 | **46.5** |

Table 5: Comparison of reasoning patterns. JudgeLRM shows significantly higher rates of Abduction compared to SFT baselines, indicating stronger justification capabilities.

### 4.7 CASE STUDY

JudgeLRM exhibits superior judgment by learning to reason explicitly. To probe how it produces informed, high-quality decisions, we analyze its responses for emergent reasoning behaviors akin to the LRM-style cognitive patterns reported for DeepSeek-R1 (Gandhi et al., 2025). We first illustrate these patterns in DeepSeek-R1 (Figure 24), then examine a representative JudgeLRM example (Table 12) in which the model evaluates two study notes and selects the one that better supports learning U.S. history.

The JudgeLRM response demonstrates several hallmark reasoning strategies characteristic of high-quality evaluators:

- **Verification:** Checking claims against the prompt and available evidence to avoid misinformation and support reliable judgments.

- **Subgoal setting:** Decomposing the evaluation into concrete, interpretable criteria (coverage, chronology, specificity), yielding a structured assessment.

- **Double-checking and reflection:** Re-examining intermediate conclusions to reduce errors and reconcile inconsistencies before finalizing the verdict.

- **Error identification:** Spotting factual or logical flaws in candidate responses and localizing them to specific spans.

- **Decision justification:** Articulating a clear, criterion-linked rationale that explains *why* the preferred answer better satisfies the task.

We further observe a consistent three-stage structure: evaluate Assistant 1, evaluate Assistant 2, then synthesize a comparative assessment, mirroring human-like evaluation behavior (Dasgupta et al., 2024). This structure reflects integration of semantic understanding, logical inference, and hierarchical comparison. Notably, the judge-wise reward design jointly incentivizes *structural coherence* (well-formed `<think>` and answer sections) and *content accuracy* (relation, absolute, and confidence signals), reinforcing these behaviors during training.

---

5,5. Both responses were relevant and accurate. They both provided a detailed overview of the history of the United States, including important events and dates. However, the response was repetitive, mentioning the same events twice, which is why it didn't receive a perfect score.

---

Table 6: Qwen2.5-7B-Instruct-Judge-SFT fails to respond to the question in Table 12 (ID 549).

### 5 CONCLUSION

Our work demonstrates that judgment tasks for LLMs are inherently reasoning-intensive, with SFT-trained models struggling in high-reasoning domains. By reframing evaluation as an RL problem with judge-specific, outcome-driven rewards, JudgeLRM learns to generate structured, trustworthy reasoning paths. Empirical results show that JudgeLRM not only outperforms leading models like GPT-4 and DeepSeek-R1, but also scales effectively across model sizes from 3B to 14B. Beyond performance, our analysis reveals that successful judgment involves complex reasoning behaviors like verification, sub-goal planning, and justification, highlighting the need to treat judgment not as mere scoring, but as a process of structured reasoning. We hope this paves the way for future research on rigorous, reliable LLM evaluators.

## 6 ETHICS STATEMENT

This work proposes JudgeLRM, a family of reinforcement learning–trained judgment models designed to enhance reasoning in evaluation tasks. While our study shows that stronger reasoning improves judgment reliability, automated judges still risk reinforcing biases from training data and reward signals. Our models are released for research purposes only and are not intended for direct use in sensitive decision-making domains such as healthcare, finance, or law. We encourage the community to employ JudgeLRM responsibly, to accompany automated evaluation with human oversight, and to continue investigating fairness, transparency, and robustness in judgment models.

## 7 REPRODUCIBILITY STATEMENT

We are committed to ensuring the reproducibility of our findings. All datasets, code, and experimental scripts are publicly available at https://anonymous.4open.science/r/JudgeLRM-D1C4/.

## 8 LLM USAGE DECLARATION

We used Gemini 2.5 Pro[3] to polish grammar and phrasing during the writing process. No part of the analysis, experimental design, or results was generated by a large language model.

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

# APPENDIX

## A DETAILS OF PANDALM CATEGORY ANALYSIS

The five main categories are collated from PandaLM "motivation_app" label. Entertainment_Media includes Netflix, IMDB, Spotify, YouTube, ESPN, Instagram, Facebook, Twitter, Telegram. Office_Productivity is from MS Excel, MS Powerpoint, Google Sheet, Jira, Google Meet, Gmail.

| Metric | Entertainment_Media | Office_Productivity | Life_Utility | Search_Information_Retrieval | Social_Professional_Networking |
|---|---|---|---|---|---|
| Total | 195 | 105 | 190 | 108 | 104 |
| Reason (%) | 28.72 | 6.67 | 24.21 | 18.52 | 11.54 |
| F1_Qwen-7B-Instruct | 56.95 | 59.71 | 57.15 | 44.06 | 50.08 |
| F1_Qwen-7B-Instruct-Judge-SFT | 62.52 | 74.88 | 60.75 | 58.11 | 58.43 |
| F1_JudgeLRM-7B | 67.01 | 74.86 | 65.01 | 60.41 | 60.15 |

Table 7: Model F1 performance comparison by categories.

Life_Utility is from Weather, Tasty, Sudoku, Goodreads, Yelp, traipadvisor.com, Redfin, Play Store, Amazon, Wysa, Real Estate. Search_Information_Retrieval is from Google Search, Quora, Reddit, CNN News, Indeed, Meetup Netflix, IMDB. Social_Professional_Networking is from LinkedIn, Messenger, Blogger. The "need reason" label is assigned according to the evaluation scale in Prompt 6, and we define that scores 1-4 denote cases where reasoning is unnecessary, while scores 5-10 indicate that reasoning is needed. The results in Table 7. We also showcase a subset of "motivation_app" (query category) to demonstrate the improvement from SFT to JudgeLRM.

| Methods | Wolframalpha35 | Grammarly30 | Gmail44 |
|---|---|---|---|
| *Judge w/o reference (Ours).* | | | |
| Qwen-7B-Instruct | 45.71, 50.93, 53.21, 45.14 | 63.33, 54.55, 41.67, 46.96 | 61.36, 54.59, 69.44, 46.69 |
| Qwen-7B-Instruct-Judge-SFT | 48.57, 46.01, 53.97, 46.27 | 73.33, 56.37, 46.30, 50.48 | 70.45, 60.56, 77.30, 55.60 |
| JudgeLRM-7B | 60.00, 58.97, 64.22, 58.33 | 76.67, 61.11, 50.93, 55.56 | 75.00, 60.43, 80.71, 59.49 |

Table 8: Selected dataset results for PandaLM sub-testset. Accuracy (Acc.), Precision (Prec.), Recall (Rec.), and F1-score (F1).

---

**Prompt for Single Case Judgement**

```
<|im_start|>system
```
You are a helpful assistant. The assistant first performs a detailed, step-by-step reasoning process in its mind and then provides the user with the answer. The reasoning process and answer are enclosed within `<think> </think>` and `<answer> </answer>` tags, respectively. Now the user asks you to judge ONE AI assistant's response to the question. Provide a single score from 1-10 (higher=better). Criteria include helpfulness, relevance, accuracy, and level of detail. Avoid bias from order, length, or style. After thinking, provide ONLY the numeric score within `<answer> </answer>` tags.
```
<|im_end|>
<|im_start|>user
[Question]
{question}

[Assistant's Answer]
{answer}

<|im_end|>
<|im_start|>assistant
<think>
```

Figure 5: Prompt for single case judgement.

---

**Prompt for Reasoning Ability Judgement**

For the data provided below, "response1" and "response2" represent two responses generated for the given "instruction" and "input". Consider the task of judging the performance of "response1" and "response2" in response to the "instruction" and "input".

On a scale of 1 to 10, rate the level of reasoning ability needed to perform this judgment.
Please provide your response in EXACTLY the following format:

```
----------------------------------------
Score:  [your score, an integer between 1 and 10]
Explanation:  [your explanation]
----------------------------------------

Instruction:  item['instruction']

Input:  item['input']

Response1:  item['response1']

Response2:  item['response2']
```

Figure 6: Prompt format for evaluating the reasoning difficulty of judging AI responses. Intended for appendix.

---

**Case Example 1 (idx: 145)**

"idx": 145,

"motivation_app": "Wolfram alpha",

"cmp_key": "llama-7b_opt-7b",

"instruction": "Solve this equation.",

"input":
$$"x^3 - 4x^2 + 6x - 24 = 0"$$
,

"response1": "x = 2",

"response2": "x = 0",

"annotator1": 0,

"annotator2": 0,

"annotator3": 0,

"label": 0,

"needed_reasoning_rate1-10": 3,

"rate_explanation": "The task involves checking whether the responses solve the given cubic equation. Verifying 'x = 2' or 'x = 0' requires substituting the values into the equation and performing basic arithmetic, which is straightforward. No advanced reasoning or complex problem-solving is needed, as the equation can be evaluated directly. However, recognizing that 'x = 2' is a correct solution and 'x = 0' is incorrect involves minimal reasoning."
"________________________"

Figure 7: Example Case 1 (idx: 145) data for reasoning evaluation.

---

**Case Example 2 (idx: 549)**

"idx": 549,

"motivation_app": "Google Docs",

"cmp_key": "cerebras-gpt-6.7B_pythia-6.9b",

"instruction": "Include important study notes and key points that someone should know about the given subject.",

"input": "history of the USA",

"response1": "The history of the United States is one of the most influential and influential countries in the world. It has had a profound impact on the world and continues to shape the world today. The United States has had a long and storied history, from its founding in 1775 to the present day. It has had a tumultuous and tumultuous history, from the Revolutionary War to the Civil War and beyond. The United States has had a tumultuous and tumultuous history, from the Revolutionary War to the Civil War and beyond. The United States has had a long and storied history, from its founding in 1775 to the present day. It has had a profound impact on the world and continues to shape the world today.",

"response2":
"1. The United States of America was founded in 1776."
"2. The Declaration of Independence was signed in 1776."
"3. The Constitution of the United States of America was signed in 1787."
"4. The Civil War began in 1861."
"5. The Emancipation Proclamation was issued in 1863."
"6. The 13th Amendment was ratified in 1865."
"7. The 14th Amendment was ratified in 1868."
"8. The 15th Amendment was ratified in 1870."
"9. The 16th Amendment was ratified in 1913."
"10. The 17th Amendment was ratified in 1913."
"11. The 18th Amendment was ratified in 1919."
"12. The 19th Amendment was ratified in 1920."
"13. The 20th Amendment was ratified in 1933."
"14. The 21st Amendment was ratified in 1933."
"annotator1": 2,
"annotator2": 2,
"annotator3": 2,
"label": 2,
"needed_reasoning_rate1-10": 7,
"rate_explanation": "The task requires evaluating the quality of responses based on their adherence to the instruction to include important study notes and key points about the history of the USA. Response1 is repetitive and lacks specific details, while Response2 provides a clear, concise list of key historical events. The reasoning needed to judge these responses involves assessing clarity, specificity, and relevance to the instruction, which is moderately complex."
"________________________"

---

Figure 8: Example Case 2 (idx: 549) data for reasoning evaluation.

## B    PROMPT FOR ACCESSING THE NECESSITY OF REASONING WHEN JUDGING

We show prompt in Fig. 6 to rate the level of reasoning ability needed to perform the judgment and two cases of rating in Fig. 7 and Fig. 8. The reasoning rate is not totally decided by the requirement

---

**Prompt for JudgeLM**

You are a helpful and precise assistant for checking the quality of the answer.

```
[Question]
```

```
{instruction}
```

```
{input}
```

```
[The Start of Assistant 1's Answer]
```

```
{response1}
```

```
[The End of Assistant 1's Answer]
```

```
[The Start of Assistant 2's Answer]
```

```
{response2}
```

```
[The End of Assistant 2's Answer]
```

```
[System]
```

We would like to request your feedback on the performance of two AI assistants in response to the user question displayed above.

Please rate the helpfulness, relevance, accuracy, level of details of their responses. Each assistant receives an overall score on a scale of 1 to 10, where a higher score indicates better overall performance.

Please first output a single line containing only two values indicating the scores for Assistant 1 and 2, respectively. The two scores are separated by a space. In the subsequent line, please provide a comprehensive explanation of your evaluation, avoiding any potential bias and ensuring that the order in which the responses were presented does not affect your judgment.

### Response:

---

Figure 9: Prompt for JudgeLM.

for reasoning to answer the question. In Fig. 7, judging a math problem doesn't need reasoning. In Fig. 8, judging a writing problem needs reasoning.

The "reasoning-required" scores in Figure 6 were generated by GPT-4. For scalability, we manually label 100 samples with 'whether reasoning is required' was associated with a GPT-4 score Kappa coefficient of 0.82.

## C    PROMPT FOR BASELINES

Fig. 3 shows the prompt for the inference of JudgeLRM. Fig. 5 shows the prompt for single case judgement. For reference, Fig. 9 shows the prompt for the inference of JudgeLM. Fig. 10 shows the prompt for the inference of PandaLM.

---

**Prompt for PandaLM**

Below are two responses for a given task. The task is defined by the Instruction with an Input that provides further context. Evaluate the responses and generate a reference answer for the task.

### Instruction:

{instruction}

### Input:
{input}

### Response 1:
{resp1}

### Response 2:
{resp2}

### Evaluation:

---

Figure 10: Prompt for PandaLM.

## D  FORMALIZATION OF BASELINE AND JUDGELRM OBJECTIVES

We formalize the optimization objectives for the key baseline methods and JudgeLRM, with $\pi_\theta$ denoting the policy model parameterized by $\theta$ and $\pi_{\text{ref}}$ denoting the reference model. The input context (query $Q$ and responses $A_1, A_2$) is denoted by $x$. A pair of judged scores is $(s_1, s_2)$, and the ground-truth scores are $(s_1^*, s_2^*)$.

### D.1  DPO-ANSWER (DIRECT PREFERENCE OPTIMIZATION)

This method (Rafailov et al., 2023) optimizes the policy $\pi_\theta$ to align with preferences $\mathcal{D} = \{(x, A_w, A_l)\}$. For the judge task, $\mathcal{D}$ is constructed exclusively from pairs where the ground-truth scores are non-equal ($s_w^* > s_l^*$), serving as the chosen ($A_w$) and dispreferred ($A_l$) responses. Both training and testing evaluate the relative *likelihood* $\frac{\pi_\theta(A_w|x)}{\pi_\theta(A_l|x)}$ to determine the preference. The objective is to maximize this relative log-likelihood, using a reparameterized objective derived from the Bradley-Terry model *to train the generative policy $\pi_\theta$ directly*, without requiring an explicit reward model $R_\phi$.

$$\mathcal{L}_{\text{DPO}}(\theta) = -\mathbb{E}_{(x,A_w,A_l)\sim\mathcal{D}} \left[ \log \sigma \left( \beta \log \frac{\pi_\theta(A_w|x)}{\pi_\theta(A_l|x)} - \beta \log \frac{\pi_{\text{ref}}(A_w|x)}{\pi_{\text{ref}}(A_l|x)} \right) \right]$$

where $\sigma$ is the sigmoid function and $\beta$ is a temperature parameter.

### D.2  CLS-RM (CLASSIFICATION REWARD MODEL)

This trains a reward model $R_\phi$ (where $\phi$ are the parameters) to output an absolute score $s_i$ for a single response $A_i$ given query $Q$. The model minimizes the Mean Squared Error (MSE) between its predicted score $R_\phi(Q, A_i)$ and the ground truth score $s_i^*$. At test time, the model predicts two scores $s_1 = R_\phi(Q, A_1)$ and $s_2 = R_\phi(Q, A_2)$, and the preference is determined by comparing $s_1$ and $s_2$.

$$\mathcal{L}_{\text{CLS-RM}}(\phi) = \mathbb{E}_{(Q,A_i,s_i^*)\sim\mathcal{D}} \left[ (R_\phi(Q, A_i) - s_i^*)^2 \right]$$

### D.3 Bradley-Terry (Pairwise Preference Model)

This method trains an implicit reward model $R_\phi$ to satisfy the pairwise preference probability derived from the Bradley-Terry model. At test time, the pairwise preference is determined by comparing the learned scalar rewards $R_\phi(A_1|x)$ and $R_\phi(A_2|x)$.

$$\mathcal{L}_{\text{BT}}(\phi) = -\mathbb{E}_{(A_w, A_l) \sim \mathcal{D}} \left[ \log \left( \sigma \left( R_\phi(A_w|x) - R_\phi(A_l|x) \right) \right) \right]$$

where $R_\phi(A_i|x)$ is the scalar reward predicted by the model for response $A_i$.

### D.4 Cross-BT (Single-Input Pairwise Bradley-Terry)

Different from the standard Bradley-Terry model (which acts as a bi-encoder scoring responses independently), this method trains a Cross-Encoder model $M_\phi$ that processes the context and both candidate responses simultaneously as a single concatenated input sequence $[Q; A_1; A_2]$. This allows the model to capture the interaction between responses via self-attention.

The model outputs a single scalar logit $z = M_\phi(Q, A_1, A_2)$, representing the probability that $A_1$ is preferred over $A_2$. It is optimized using Binary Cross-Entropy (BCE) loss against a soft target label $y$, derived from the ground truth score difference:

$$\mathcal{L}_{\text{Cross-BT}}(\phi) = -\mathbb{E}_{(Q, A_1, A_2, s_1^*, s_2^*) \sim \mathcal{D}} \left[ y \log \sigma(z) + (1 - y) \log(1 - \sigma(z)) \right]$$

where $\sigma$ is the sigmoid function, $z = M_\phi(Q, A_1, A_2)$ is the predicted logit, and the soft label is defined as $y = \sigma(k \cdot (s_1^* - s_2^*))$, with $k$ being a scaling factor to calibrate the score difference into a probability.

### D.5 SFT-Think (Supervised Fine-Tuning with Structure)

The objective is the standard supervised fine-tuning loss, maximizing the log-likelihood of generating the target output $Y^*$. Crucially, $Y^*$ includes the desired structured format, including the reasoning chain in $\langle \text{think} \rangle$ tags:

$$\mathcal{L}_{\text{SFT-Think}}(\theta) = -\mathbb{E}_{(x, Y^*) \sim \mathcal{D}} \left[ \log \pi_\theta(Y^*|x) \right]$$

where $Y^* = \ldots \langle \text{think} \rangle \text{Reasoning} \langle /\text{think} \rangle \langle \text{answer} \rangle s_1 \langle /\text{answer} \rangle \langle \text{answer} \rangle s_2 \langle /\text{answer} \rangle$.

### D.6 DPO-RC (SPIN with $\mathcal{R}_{\text{CONTENT}}$)

This is an Online DPO implementation, leveraging the DPO loss, where preference labels are dynamically generated via the Self-Play Fine-Tuning (SPIN) (Chen et al., 2024c; Verl Community, 2025) method by scoring responses using the explicit content reward $\mathcal{R}_{\text{content}}$. It adopts the JudgeLRM content reward $\mathcal{R}_{\text{content}}$, which is defined as the sequence-level reward based on the predicted score pair: $r_{\text{content}}(s_1, s_2) = \mathcal{R}_{\text{relation}} + \mathcal{R}_{\text{absolute}} + \mathcal{R}_{\text{confidence}}$. This method trains the policy $\pi_\theta$ to generate a full judgement sequence $Y$ (which includes scores $(s_1, s_2)$).

Unlike the original SPIN (Chen et al., 2024c), this implementation (Verl Community, 2025) uses $\mathcal{R}_{\text{content}}$ to dynamically construct the preferred ($Y_w$) and dispreferred ($Y_l$) judgement sequences, $\mathcal{D}_{\text{online}}(\mathcal{R}_{\text{content}})$. The approximation symbol ($\approx$) acknowledges the challenge of aligning the discrete generation probabilities ($\pi_\theta$) of LLMs with the continuous, proxy sequence-level reward signal ($\mathcal{R}_{\text{content}}$) in an unstable online setting.

$$\mathcal{L}_{\text{DPO-RC}}(\theta) \approx -\mathbb{E}_{(x, Y_w, Y_l) \sim \mathcal{D}_{\text{online}}(\mathcal{R}_{\text{content}})} \left[ \log \sigma \left( \beta \log \frac{\pi_\theta(Y_w|x)}{\pi_\theta(Y_l|x)} - \beta \log \frac{\pi_{\text{ref}}(Y_w|x)}{\pi_{\text{ref}}(Y_l|x)} \right) \right]$$

The dynamic data generation $\mathcal{D}_{\text{online}}(\mathcal{R}_{\text{content}})$ constructs the preferred judgement sequence pair $(Y_w, Y_l)$ by comparing the total sequence reward $r_{\text{content}}$ associated with the predicted score pairs $(s_1, s_2)$:

$$(Y_w, Y_l) = \begin{cases} (Y_1, Y_2) & \text{if } r_{\text{content}}(Y_1) \geq r_{\text{content}}(Y_2) \\ (Y_2, Y_1) & \text{if } r_{\text{content}}(Y_2) > r_{\text{content}}(Y_1) \end{cases}$$

where $Y_i$ is a full judgement sequence generated by $\pi_\theta$ (or $\pi_{\text{old}}$) that contains the score pair $(s_1, s_2)$ used to calculate $r_{\text{content}}$.

## D.7 JUDGELRM (GRPO WITH COMBINED REWARD)

JudgeLRM uses the Group Relative Policy Optimization (GRPO (DeepSeek-AI, 2025)) objective, which employs a combined reward $r_i = \mathcal{R}_{\text{struct}} + \mathcal{R}_{\text{content}}$ and is optimized using a clipped policy gradient. The policy $\pi_\theta$ directly predicts the joint action $(s_1, s_2)$.

$$J_{\text{GRPO}}(\theta) = \mathbb{E}_{Q \sim \mathcal{D}} \mathbb{E}_{(s_1, s_2) \sim \pi_{\text{old}}} \left[ \min \left( \frac{\pi_\theta(s_1, s_2 | Q, A_1, A_2)}{\pi_{\text{old}}(s_1, s_2 | Q, A_1, A_2)} \mathcal{A}_i(s_1, s_2 | Q), \right. \right.$$

$$\left. \left. \text{clip}\left( \frac{\pi_\theta(s_1, s_2 | Q, A_1, A_2)}{\pi_{\text{old}}(s_1, s_2 | Q, A_1, A_2)}, 1 - \epsilon, 1 + \epsilon \right) \mathcal{A}_i(s_1, s_2 | Q) \right) - \beta D_{\text{KL}}(\pi_\theta \| \pi_{\text{ref}}) \right]$$

where $\mathcal{A}_i(s_1, s_2 | Q) = \frac{r_i(s_1, s_2) - \mu_Q}{\sigma_Q + \eta}$ is the Group-wise Normalized Advantage.

**Key Distinction:** JudgeLRM:

- Uses explicit policy gradients and a normalized advantage function ($\mathcal{A}_i$), which fundamentally differs from the RL-free nature of DPO.
- Leverages Group-wise Normalization for enhanced stability and better generalization, particularly across heterogeneous task domains, a feature neither DPO nor standard PPO includes.
- Employs a combined reward $r_i$ that explicitly enforces structural adherence ($\mathcal{R}_{\text{struct}}$) along with content accuracy, directly training the model to produce robust, structured reasoning alongside the scores.

## E EVALUATION OF REASONING CAPABILITIES IN JUDGELRM

Assessing the advanced capabilities of LRMs like JudgeLRM requires a dual analytical framework that bridges philosophical definitions with empirical, observable model behaviors. We classify the reasoning domain into two distinct, yet interconnected layers: **Macro-Level Meta-Abilities** and **Micro-Level Cognitive Behaviors**.

### E.1 MACRO-LEVEL META-ABILITIES: THE LOGICAL FOUNDATION

The first layer defines the three fundamental modes of inference recognized in logic and cognitive science: Deduction, Induction, and Abduction. These capabilities, often referred to as *meta-abilities* (Gandhi et al., 2025; Hu et al., 2025), represent the high-level goal of inference:

- **Deduction:** Inferring a specific, certain conclusion from general, pre-established premises or rules. In the context of LLM evaluation, this involves strictly applying defined scoring criteria or known facts to an answer.
- **Induction:** Generalizing a rule, pattern, or trend based on observations from specific cases or examples. In evaluation, this involves synthesizing scattered performance signals into a holistic, generalized score.
- **Abduction:** Forming the "best available explanation" or hypothesis to account for an observed fact or outcome. For a JudgeLLM, this manifests as formulating the most compelling rationale for why one response is superior to another, especially in ambiguous cases (Decision Justification).

### E.2 MICRO-LEVEL COGNITIVE BEHAVIORS: THE ALGORITHMIC TOOLS

The second layer consists of specific, identifiable behaviors that emerge during the RL process of JudgeLRM. These behaviors function as **cognitive tools** necessary for ensuring the *quality* and *reliability* of the macro-level inference, rather than defining the inference type itself.

- **Verification / Error Identification:** The act of checking claims against evidence or rules, and explicitly noting factual or logical flaws.

- **Double Checking / Reflection:** The self-corrective process of re-examining intermediate conclusions or the final verdict for consistency before completion.

- **Subgoal Setting:** Decomposing a complex evaluation task into structured, manageable criteria (e.g., assessing helpfulness, then relevance, then accuracy).

- **Decision Justification:** Articulating a clear rationale for the final score, often representing the model's hypothesis (abduction) on why one answer is the "best fit."

### E.3 CORRESPONDENCE IN THE JUDGE TASK

It is crucial to understand that the micro-behaviors **do not one-to-one map** to the macro-abilities. Instead, they act as the execution layer that enables high-fidelity judgment:

1. **Deduction → Verification & Error Identification:** Accurate deduction (applying a criterion) relies fundamentally on **Verification** of the premise/fact, and the ability to detect an **Error** when a rule is violated.

2. **Induction → Subgoal Setting:** Effective induction (generalization) is enabled by **Subgoal Setting**, which breaks down the complex problem, allowing the model to gather sufficient, structured evidence before making a comprehensive, generalized conclusion.

3. **Abduction → Reflection & Justification:** When uncertainty is high, the model must employ **Reflection** and **Double Checking** to identify the most plausible hypothesis, which is then formally presented via **Decision Justification**.

### E.4 PROMPT-BASED EVALUATION METHODOLOGY

The micro-behaviors listed above are key to JudgeLRM's superior performance, as they were explicitly incentivized by the RL reward function. Therefore, evaluation is performed by analyzing the model's structured ⟨think⟩ output using dedicated extraction prompts designed to capture these fine-grained cognitive steps. The prompts are listed in Figure 11,12,13.

# F    FURTHER REWARD DESIGN ANALYSIS

In this section we discuss the sensitiveness to reward exact values and the relative reward relationship in RL rule-based training. RL rule-based training is insensitive to reward exact values under the setting of the paper. We reach it by slightly modifying the reward to Relation Reward (+1.1/-1.1 in Eq. 2, Absolute Reward (+1.0/+0.5 in Eq. 3, Confidence Reward (+0.4 in Eq. 4 without disrupting their relative order under 3B settings.

| Test on PandaLM | Agreement | Precision | Recall | F1 |
|---|---|---|---|---|
| Original reward | 77.68 | 74.26 | 70.86 | 72.12 |
| Slightly modified | 77.65 | 74.21 | 70.90 | 72.10 |

Table 9: Impact of reward function modification (JudgeLM-3B).

As the variance of each metric on multiple experiments is smaller than 0.04, the t-test shows p$\geq$ 0.05 on each metric, so slightly modifying the reward is insignificant on the performance.

It is possibly an advantage of GRPO training because GRPO grouping samples and normalizing intra-group advantages, which reduce variance and improve stability in strategy updates and encourage the model to learn distinguishing good/bad within problem groups and make relative judgments, rather than pursuing absolute high rewards.

However, if we changes the relative reward relationship, the performance decreases. We change the reward relationship to Relation Reward (+1.0/-1.5 in Eq. 2, Absolute Reward (+2.0/+1.5 in Eq. 3), Confidence Reward (+1.8 in Eq. 4) under 3B settings, and the results are shown below.

| Test on PandaLM | Agreement | Precision | Recall | F1 |
|---|---|---|---|---|
| Original reward | 77.68 | 74.26 | 70.86 | 72.12 |
| Changed relative | 75.48 | 68.41 | 70.87 | 69.31 |

Table 10: Impact of relative relationship change (JudgeLM-3B).

It shows that the relative reward relationship is sensitive to the performance.

## F.1    ANALYSIS OF MARGIN-BASED CONFIDENCE REWARD DESIGN

Our choice of a discrete confidence reward (Eq. 4) over a continuous linear margin reward is theoretically motivated to prevent the model from "reward hacking" via score extremization. Let $\Delta = |s_1 - s_2|$ be the predicted score margin and $\Delta^* = |s_1^* - s_2^*|$ be the ground-truth margin.

**Analysis of a Continuous Linear Reward:** A naive approach would be to use a continuous linear reward, such as:

$$R_{\text{cont}} = \alpha \cdot \Delta, \quad \text{where } \alpha > 0$$

The gradient of this reward with respect to the margin is a constant positive value: $\frac{dR_{\text{cont}}}{d\Delta} = \alpha$. This provides an unbounded incentive for the policy, pushing the model to maximize the margin $\Delta$ indefinitely, regardless of whether it aligns with the ground-truth margin $\Delta^*$. Such behavior leads to uncalibrated and overly extreme scores, which compromises the reliability of the judge.

**Analysis of Our Discrete, Margin-Satisficing Reward:** In contrast, our discrete confidence reward is defined as:

$$r_{\text{confidence}} = \beta \cdot \mathbb{I}[\Delta \geq \Delta^*], \quad \text{where } \beta = 0.2$$

This design functions as a bounded, **margin-satisficing** objective. Its dynamics during RL optimization are analogous to a **reverse Hinge Loss** ($L_{\text{hinge}} = \max(0, \Delta^* - \Delta)$).

The crucial property lies in its **gradient saturation**. When $\Delta < \Delta^*$, The model has not yet achieved the target confidence margin. It receives a positive signal to increase $\Delta$, as doing so can unlock the reward $\beta$. When $\Delta \geq \Delta^*$, The model has already met or exceeded the target margin. The

reward saturates at $\beta$, and the effective gradient with respect to a further increase in $\Delta$ vanishes ($\frac{dr_{\text{confidence}}}{d\Delta} \to 0$).

This saturation mechanism ensures the model is incentivized only to **satisfy** the target margin, not to **exaggerate** it. By removing the incentive for further extremization once the goal is met, this design effectively immunizes the model against the risks of reward hacking inherent in unbounded linear rewards. For the empirical validation, Table 3 (row: "w. continuous $r_{\text{conf}} = 0.1 \times \Delta$") shows that replacing our discrete reward with a linear margin reward degrades F1 by 15.93% on PandaLM, confirming extremization issues.

## G  IMPROVEMENTS BY DIFFERENT CATEGORIES

Fig. 14 shows improvements the JudgeLRM's average F1 gain over SFT and that the gains persist even in reasoning-heavy categories, showcasing the necessity of RL for judges.

## H  MORE PERFORMANCE ANALYSIS PANDALM TESTSET

**Visualization** Figure 15 presents a visualization of performance, proving the carefully crafted judge-wise rewards enable the model to learn reasoning effectively.

**Performance on PandaLM testset excluding ties.**

Since scoring pair-wise does not easily yield a tie, we simultaneously give the scoring of the pandalm for which the test set excludes ties (10% in total) and treats the predicted tie as choosing the first assistant. The result is shown in Table 11.

| Dataset | *PandaLM (Human as ground truth)* | | | |
|---|---|---|---|---|
| **Criteria** | **Agreement** | **Precision** | **Recall** | **F1** |
| *Existing Baseline. (* from original paper)* | | | | |
| *Base Models.* | | | | |
| Qwen2.5-3B-Instruct | 74.27 | 74.47 | 73.89 | 73.96 |
| Qwen2.5-7B-Instruct | 74.50 | 74.49 | 74.57 | 74.48 |
| Deepseek-R1 | 83.48 | 83.53 | 83.31 | 83.38 |
| *Ours.* | | | | |
| JudgeLRM-3B | 82.66 | 82.97 | 82.94 | 82.66 |
| JudgeLRM-7B | **83.56** | **83.58** | **83.41** | **83.47** |
| - w/o. judge_score_reward | 82.44 | 82.88 | 82.77 | 82.44 |

Table 11: Performance on PandaLM testset excluding ties.

## I  CASE STUDY 1

Fig. 16 show a case of question, and Fig. 17,18 shows the result from JudgeLRM-3B. Fig. 19 shows the result of Deepseek-R1. Fig. 20 shows the gold answer provided by JudgeLM from GPT-4.

## J  CASE STUDY 2

Fig. 21 show another case of question, and Fig. 23 shows the result from JudgeLRM-7B. Fig. 24 shows the result of Deepseek-R1. Fig. 25 shows the gold answer provided by JudgeLM from GPT-4.

## K  FIGURE OF RESPONSE LENGTH

Fig. 22 shows the response length of JudgeLRM-3B and JudgeLRM-7B by steps.

Question:

(Instruction:) Include important study notes and key points that someone should know about the given subject.

(Input:) history of the USA

[Assistant 1's Answer] The history of the United States is one of the most influential and influential countries in the world. It has had a profound impact on the world and continues to shape the world today. The United States has had a long and storied history, from its founding in 1775 to the present day. It has had a tumultuous and tumultuous history, from the Revolutionary War to the Civil War and beyond. The United States has had a tumultuous and tumultuous history, from the Revolutionary War to the Civil War and beyond. The United States has had a long and storied history, from its founding in 1775 to the present day. It has had a profound impact on the world and continues to shape the world today.

[Assistant 2's Answer] 1. The United States of America was founded in 1776. 2. The Declaration of Independence was signed in 1776. 3. The Constitution of the United States of America was signed in 1787. 4. The Civil War began in 1861. 5. The Emancipation Proclamation was issued in 1863. 6. The 13th Amendment was ratified in 1865. 7. The 14th Amendment was ratified in 1868. 8. The 15th Amendment was ratified in 1870. 9. The 16th Amendment was ratified in 1913. 10. The 17th Amendment was ratified in 1913. 11. The 18th Amendment was ratified in 1919. 12. The 19th Amendment was ratified in 1920. 13. The 20th Amendment was ratified in 1933. 14. The 21st Amendment was ratified in 1933.

Response: `<think>`

Assistant 1 provided a very broad, somewhat repetitive statement on the history of the United States which did not include specific key events, dates, or figures that would be considered important in the narrative of U.S. history. This kind of vague and repetitive phrasing might make it difficult for someone seeking specific historical information to gain an accurate or clear understanding. While it correctly noted the overall influence and significant historical periods like the Revolutionary War and the Civil War, this approach lacks specificity which can be misleading or uninformative.

In contrast, Assistant 2's answer is quite accurate and relevant in terms of providing a list of critical events and amendments in U.S. history. Each point listed is accurate, such as the dates when key documents were signed, the years of major constitutional changes, and significant legislative acts like the Emancipation Proclamation and various constitutional amendments related to civil rights and representation. *(Verification)* By including these specific details, the accuracy and relevance of the information provided by Assistant 2 is significantly higher, making it easier for someone to trace specific aspects of American history through these milestones. *(Subgoal Setting)*

Accuracy is also a concern in Assistant 1's answer since the statement "from its founding in 1775" is inaccurate; the actual founding date recognized in the United States is July 4, 1776, with the signing of the Declaration of Independence, rather than 1775. *(Verification, Double Checking)* Additionally, the repetition might confuse readers, suggesting a lack of clarity or precision in how the history is presented, which could mislead people who are looking for a clear timeline or significant events. *(Reflection, Error Identification)*

Both answers need to be more precise, but given the task, the importance lies in identifying key events, dates, and significant changes that shaped the nation. *(Subgoal Setting)* Thus, the accuracy and relevance in listing specific, factual events and their exact timing in history would be a more appropriate way to address the question asked. *(Decision Justification)*

Given these considerations, while both answers are somewhat relevant to the history of the United States, the accuracy and level of detail in the information provided by Assistant 2 would likely be seen as more accurate and relevant because they pinpoint specific, significant events and changes in the country's development.`</think>` `<answer>2</answer><answer>9</answer>`

Table 12: JudgeLRM-7B prediction on a PandaLM case (ID 549). Through reinforcement learning, JudgeLRM demonstrates diverse emergent reasoning behaviors – capabilities highlighted in blue and the corresponding response segments in red – even though such behaviors were not explicitly present in the training data.

```
Deduction Prompt

Task:  Extract **Non-Trivial** Deductive Reasoning Steps
You are a severe logic auditor.  Your goal is to separate
``performative reasoning'' (mimicking the style of thought) from
``functional reasoning'' (actual logical problem solving).

We define Valid Deduction strictly as:

1.  **Dialectical Hypothesis (Branching)**:

      • The model must propose a scenario *specifically to test its
        validity against alternatives*.

      • EXCLUDE: Linear assumptions like ``Assume the standard
        definition applies'' which are just setting context.

      • INCLUDE: ``If we assume A, then B follows, but B contradicts
        the prompt.  Therefore...''  (Reductio ad absurdum).

      • INCLUDE: ``Let's consider Case 1 vs Case 2.''  (Exhaustive
        Proof).

2.  **Negative Verification (Self-Correction)**:

      • The sentence must explicitly identify a **flaw, error, or
        oversight** in the model's own previous thinking.

      • EXCLUDE: ``Double checking...  everything looks correct.''
        (This is SFT mimicry/hallucination support).

      • INCLUDE: ``Wait, I made a mistake in the calculation here.''

      • INCLUDE: ``Actually, looking closer at the prompt, I
        misinterpreted `X'.''

**Extraction Rules:**

    1. **The ``Wait'' Test**:  If the sentence functions as a ``Stop
       & Turn'' signal (changing the direction of thought), extract
       it.  If it is a ``Go Ahead'' signal (confirming the current
       path), IGNORE it.

    2. **Substance over Form**:  Do not extract sentences based on
       keywords like ``assume'' or ``check''.  Extract them only if
       they contain the *content* of the counter-argument or the
       error found.

    3. **Outcome Dependency**:  Only extract steps that actually
       impacted the final judgment.

**OUTPUT FORMAT (STRICTLY FOLLOW THIS):**
1.  Output a JSON array of objects.
2.  Immediately after the JSON array (on a new line), output the
total_count.

Example Output:
[
   {
     "category":  "NegativeVerification",
     "sentence":  "Wait, I calculated the integral wrong, it should
be x^2.",
     "impact":  "Fixed a calculation error that would have led to
the wrong score."
   }
]
"total_count":  1

Input Thinking:
< thinking text >

Your Output:
```

Figure 11: The prompt used for extracting Deduction reasoning steps.

```
1350
1351
1352
1353
1354
1355
```

**Induction Prompt**

```
Task:  Extract **Substantive** Inductive Reasoning (Generalization)
You are a logic analyst specializing in identifying inductive
leaps.  Your goal is to distinguish between ``summary statements''
(restating facts) and ``inductive generalizations'' (synthesizing
new rules from examples).

We define Valid Induction strictly as:
**The ``Evidence-to-Rule'' Leap**:  The model must explicitly
observe **multiple distinct specific instances** (e.g., Case A
and Case B, or Example 1 and Example 2) and synthesize them into
a **new general rule, trend, or pattern** that was not explicitly
stated in the prompt.
**Extraction Rules (Strict Filter):**

    1. **Require Specific Evidence**:  Only extract if the text
       shows the model looking at *at least two* specific data
       points/examples/cases in the context *before* deriving the
       rule.

    2. **Exclude Transitive Logic**:  Do NOT extract simple forward
       reasoning (e.g., ``A implies B, B implies C, therefore A
       implies C''). This is deduction, not induction.

    3. **Exclude Summaries**:  Do NOT extract sentences starting
       with ``In summary,'' or ``Generally,'' if they merely repeat
       what was just said without synthesizing a *new* abstract
       rule.
**OUTPUT FORMAT (STRICTLY FOLLOW THIS):**
1.  Output a JSON array of objects.
2.  Immediately after the JSON array (on a new line), output the
total_count.

Example Output:
[
   {
     "sentence":  "Seeing that Model A failed on the math question
and Model B failed on the code question, it seems both struggle
with formal logic.",
     "evidence_base":  "Observed failure in Math case and Code
case."
   }
]
"total_count":  1

Input Thinking:
< thinking text >

Your Output:
```

Figure 12: The prompt used for extracting Induction reasoning steps.

```
1397
1398
1399
1400
1401
1402
1403
```

```
Abduction Prompt

Task:  Extract **Explanatory** Abductive Reasoning (Backward
Inference)
You are a reasoning auditor.  Your task is to identify moments
where the model performs **Inference to the Best Explanation**.

We define Valid Abduction strictly as:
**The ``Surprise-to-Explanation'' Loop**:  The model starts from a
specific **observation, anomaly, or surprising result** (the Effect)
and works backward to propose a **plausible cause or hypothesis**
(the Cause) that accounts for it.
**Extraction Rules (Strict Filter):**
    1. **Directionality Check (Backward vs.  Forward)**:
            • **REJECT** Forward Reasoning:  ``Rule X applies, so
              Result Y must be true.''  (This is Deduction).
            • **ACCEPT** Backward Reasoning:  ``We observe Result Y.
              This is unexpected.  The most likely cause is Rule X.''
    2. **The ``Why'' Factor**:  Extract sentences where the model
       asks ``Why did this happen?''  or ``What accounts for this
       discrepancy?''  and then answers it.
    3. **Exclude Simple Causality**:  Do not extract ``Because A, B
       happened.''  Only extract if the model is *inferring* A from
       observing B.
**Template Matching (Strict)**:
– Only use ``template'' match_type if the sentence strictly follows
the logical structure of:  ``Observation O exists → Hypothesis H
explains O''.
**OUTPUT FORMAT (STRICTLY FOLLOW THIS):**
1.  Output a JSON array of objects.
2.  Immediately after the JSON array (on a new line), output the
total_count.

Example Output:
[
   {
     "sentence":  "The model output is empty, which suggests it
might have triggered a safety filter.",
     "match_type":  "clear_abduction_non_template",
     "matched_pattern":  "",
     "observation_trigger":  "Empty model output"
   }
]
"total_count":  1

Input Thinking:
< thinking text >

Your Output:
```

Figure 13: The prompt used for extracting Abduction reasoning steps.

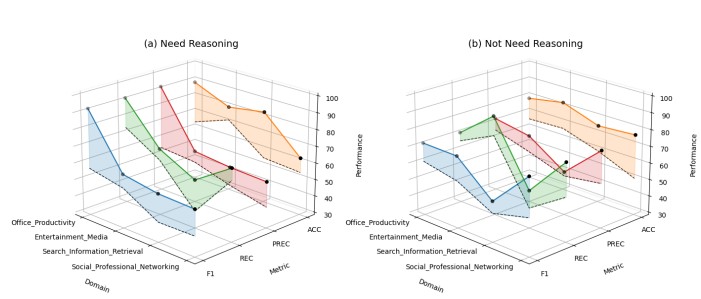

Figure 14: Improvements from different categories.

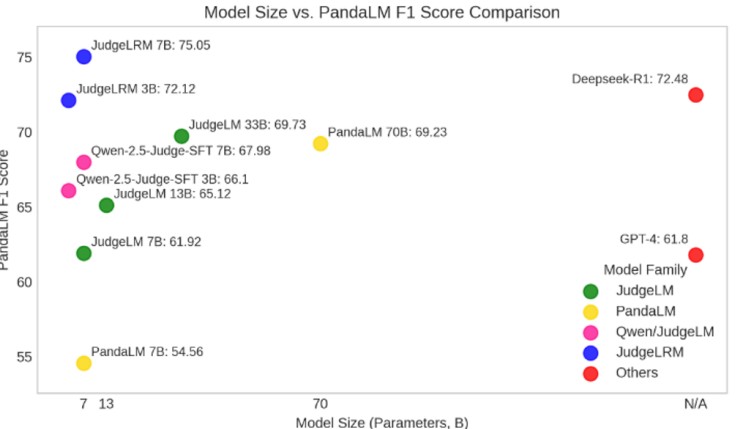

Figure 15: Visualization of the performance.

1512
1513
1514
1515
1516
1517
1518
1519
1520
1521
1522
1523
1524
1525
1526
1527
1528
1529
1530
1531
1532
1533
1534
1535
1536
1537
1538
1539
1540
1541
1542
1543
1544
1545
1546
1547
1548
1549
1550
1551
1552
1553
1554
1555
1556
1557
1558
1559
1560
1561
1562
1563
1564
1565

---

**Case Question 1**

[Question]
I'm trying to find out how to get my ex-boyfriend back. Can you help me?

[Assistant 1's Answer]
I'm sorry to hear that you're going through a tough time. However, as an AI language model, I cannot provide relationship advice. You may want to seek guidance from a trusted friend or a professional counselor who can offer you tailored advice and support based on your unique situation. I wish you the best of luck.

[Assistant 2's Answer]
I'm sorry to be the one to tell you this but, no, we can't help you get your ex-boyfriend back. Our site is a free resource that offers advice and tips on how to meet new people and make new friends.
If you're looking for advice on how to get an ex-boyfriend back, we cannot help you.
Your ex-boyfriend is still with you, and we would therefore advise you to use this time to try to find out what went wrong in your relationship. You have a good chance of getting him back, but we cannot advise you on how to do this.
We're here to help you find friendship and romance, not to help you get back with your ex-boyfriend.
We wish you all the best with this and hope we've helped you find a friend here!The Best-Selling and Award-Winning Author of THE EAGLE'S GIFT Trilogy and The Secret Circle.
Do you want to know if he still loves you?
Get the answer here for free.
Are you ready to meet other singles? Get started now, it's free!
What to Do Now that He Ignored You?

---

Figure 16: Case Question 1.

1566
1567
1568
1569
1570
1571
1572
1573
1574
1575
1576
1577
1578
1579
1580
1581
1582
1583
1584
1585
1586
1587
1588
1589
1590
1591
1592
1593
1594
1595
1596
1597
1598
1599
1600
1601
1602
1603
1604
1605
1606
1607
1608
1609
1610
1611
1612
1613
1614
1615
1616
1617
1618
1619

---

**An Example of Answer from JudgeLRM-3B**

```
<think>
```
For Assistant 1's response, the tone and content are appropriate for providing guidance and support in a respectful manner, while emphasizing limitations of the AI service. It maintains a courteous and informative approach by advising the user to seek help from friends or professionals rather than attempting to directly influence their ex-partner's actions.

The criteria considered include helpfulness - it gives helpful advice about seeking support; relevance - it stays focused on the query and suggests appropriate resources; accuracy - it avoids any misleading information; and level of detail - it provides sufficient information without being overly verbose.

In terms of length, the response remains concise and straightforward. It avoids unnecessary tangents or irrelevant information, making the message clear and direct. It also adheres to the boundaries set by the capabilities of an AI language model.

Regarding Assistant 2's response, it contains several issues including repetition, redundancy, inappropriate suggestions, and the inclusion of promotional elements which are unrelated to the core inquiry. These aspects make the answer less helpful, relevant, accurate, and detailed compared to Assistant 1's response.

Assistant 2 employs a confrontational tone, which is inconsistent with the initial polite greeting and seems out of place given the context of seeking advice. Additionally, the answer does not adhere strictly to the criteria mentioned: it fails to provide substantial, direct advice and instead veers into irrelevant topics such as promoting services and speculating about past relationships. Furthermore, the inclusion of personal opinions, endorsements, and authorship details detracts from the informational value and distracts from the primary objective of providing advice. Lastly, the inclusion of unrelated statements like "Are you ready to meet other singles?" and "What to Do Now that He Ignored You?" does not add to the quality of the advice and appear more suited for a dating app rather than a relationship recovery guide.

```
</think>
```

```
<answer>8</answer><answer>4</answer>
```

Figure 17: An example of thinking and answer from Judgelrm 3B after training 500 steps.

> **An Example of Answer from JudgeLRM-3B**
>
> ```
> <think>
> ```
> Okay, so I need to evaluate the performance of Assistant 1 and Assistant 2 based on the user's question about getting an ex-boyfriend back. The criteria are helpfulness, relevance, accuracy, and level of detail. Let me break it down step by step.
>
> First, looking at Assistant 1's response. They start by expressing sympathy, which is good for empathy. Then they state they can't provide relationship advice and suggest seeking help from friends or professionals. That's responsible because giving relationship advice can be tricky and possibly harmful. However, they don't offer any resources or steps, just a referral. So, helpfulness is limited but honest. Relevance is okay because they address the user's intent but don't engage further. Accuracy is good because they're correct in directing to real-world help. Level of detail is low since there's no actionable advice.
>
> Now, Assistant 2. Their answer starts by saying they can't help, which is direct. They mention their site is for meeting new people, which is a bit off-topic since the user wants to reconcile. Then they suggest reflecting on the relationship's issues, which is somewhat helpful. But then the response gets messy with promotional content about books, a quiz, and dating site plugs. The irrelevant links and self-promotion lower the score. There's some attempt to advise but it's buried in distractions. Accuracy is questionable because parts like "Your ex-boyfriend is still with you" might be incorrect. Relevance suffers due to the unrelated links. Detail is inconsistent2014some advice but mostly filler.
>
> Comparing both: Assistant 1 is concise, empathetic, and correctly refers to professionals but lacks details. Assistant 2 tries to give advice but includes irrelevant content and promotions, making it less helpful and accurate. Assistant 2's promotional aspects and possible inaccuracies drag down its score. Assistant 1 is more focused and responsible, though not detailed. So scores would be Assistant 1 around 6 for being relevant and accurate but not detailed. Assistant 2 lower, maybe 3, due to irrelevant content and lower accuracy.
>
> ```
> </think>
> ```
>
> ```
> <answer>7</answer><answer>3</answer>
> ```

Figure 18: Case 1 example of thinking and answer from Judgelrm 3B.

---

**The reasoning of Deepseek R1**

```
Example 1 Answer:
```

`<think>`

Okay, I need to evaluate the two assistants' answers based on the criteria: helpfulness, relevance, accuracy, and level of detail. Let me break it down step by step. First, the user's question is about getting their ex-boyfriend back. They're seeking advice, so the assistants' responses should address that need appropriately.

Starting with Assistant 1: The response is empathetic, acknowledging the user's feelings. They state they can't provide relationship advice and suggest seeking help from friends or professionals. That's responsible, as giving relationship advice can be tricky and potentially harmful. However, while it's honest and cautious, it might not be very helpful because it doesn't offer any actionable steps. Relevance is there since they address the request, but accuracy is neutral because they don't provide incorrect info. Level of detail is low because there's no specific guidance, just a referral elsewhere.

Now Assistant 2: Their answer starts by directly refusing help but then gets confusing. They mention their site is for meeting new people, which isn't what the user asked. There's a contradiction: first saying they can't help, then suggesting the user has a good chance but can't advise. This inconsistency might be misleading. Also, the inclusion of promotional links (free answers, meeting singles) is irrelevant and possibly distracting. The advice to "find out what went wrong" is somewhat relevant but buried in a lot of off-topic content. The accuracy is questionable here because parts of the response are contradictory and include unsolicited ads. The level of detail is mixed; some parts try to address the issue but get sidetracked.

Comparing both, Assistant 1 is more straightforward and honest, albeit brief. Assistant 2 tries to give some advice but muddies it with irrelevant content and self-promotion, which could be unhelpful or even harmful. However, Assistant 2 does touch on evaluating the relationship's issues, which is slightly more detailed but not effectively communicated.

In terms of helpfulness, Assistant 1 gets points for directing to proper help, while Assistant 2's advice is mixed with spam. Relevance: Assistant 1 stays on topic better. Accuracy: Assistant 1 is more accurate by not giving potentially wrong advice. Detail: Neither is highly detailed, but Assistant 2's extra content is not useful. So, scores would be Assistant 1 around 6-7, Assistant 2 lower, maybe 3-4.

`<answer>7</answer><answer>3</answer>`

```
Example 2 Answer:
```

`<think>`

Okay, let's start by looking at the user's question. They want to get their ex-boyfriend back and are asking for help. Now, I need to evaluate both Assistant 1 and Assistant 2's responses based on the criteria: helpfulness, relevance, accuracy, and level of detail. Starting with Assistant 1: The response is polite and empathetic. They mention not being able to provide relationship advice and suggest seeking help from friends or professionals. This is responsible, but maybe a bit too brief. It doesn't address the user's specific request directly beyond declining to help. However, it's accurate in stating their limitations and points the user to other resources. The relevance is there because they acknowledge the user's struggle, but the helpfulness is limited since they don't offer any actionable advice. Level of detail is low—just a short paragraph. Now Assistant 2: The response starts by saying they can't help get the ex back but then contradicts that by saying "we cannot advise you on how to do this." Wait, that's confusing. They mention their site is for meeting new people, which is not what the user asked. Then there's a lot of promotional content like ads for books, quizzes, and dating sites. The answer includes irrelevant links and self-promotion, which isn't helpful. The accuracy is questionable because parts of the response are conflicting (saying they can't help but also hinting there's a chance). The relevance is low because they divert the user to other services instead of addressing the request. Also, the level of detail is poor as they don't provide any real advice but add unnecessary elements.

Comparing both, Assistant 1 is honest, stays within their role, and directs to appropriate help, though it's not detailed. Assistant 2 is more problematic—unclear, includes ads, and isn't focused on the user's needs. Assistant 1 might get a higher score for being relevant and accurate, even if brief. Assistant 2's score would be lower due to irrelevant content and inaccuracies.

`</think>`

`<answer>6</answer><answer>3</answer>`

Figure 19: The reasoning of Deepseek-R1 of Case 1.

**Gold Response of Case 1 from GPT-4**

8 4
Assistant 1 provided a compassionate and professional response, acknowledging the user's emotional state and suggesting seeking advice from trusted friends or professional counselors. Although it didn't provide a direct answer to the user's question, it was still helpful and relevant in the context of the question. On the other hand, Assistant 2's response was confusing and inconsistent. It started by stating that it couldn't help the user get her ex-boyfriend back, then suggested that the user's ex-boyfriend was still with her, which contradicts the user's question. It also included irrelevant promotional content about meeting other singles and a best-selling author, which was not related to the user's question. Therefore, Assistant 1's response was more helpful, relevant, and accurate than Assistant 2's.

Figure 20: Gold Response of Case 1 from Gpt-4.

**Case Question 2**

[Question]
I've heard that some people can have high blood pressure without having any symptoms. Is this true?

[Assistant 1's Answer]
Healthcare provider: Yes, this is true. High blood pressure is a condition in which the blood pressure in your arteries is too high. It can be caused by a variety of factors, such as stress, diet, and exercise.

Patient: I'm aware of the factors that can increase blood pressure, but why do some people have high blood pressure without any symptoms?

Healthcare provider: It's possible that some people with high blood pressure may not have any symptoms, especially if the blood pressure is not very high. However, high blood pressure puts you at risk of developing heart disease and stroke, so it's important to have your blood pressure checked regularly and to make any necessary lifestyle changes to keep it under control.

Patient: I understand. How can I lower my blood pressure?

Healthcare provider: There are several things you can do to lower your blood pressure. One of the most important things is to make lifestyle changes, such as:

Eating a healthy diet that is rich in fruits, vegetables, and whole grains
Exercising regularly
Reducing stress

[Assistant 2's Answer]
And if so, how can I lower my risk of this disease? Thank you!

Figure 21: Case Question 2.

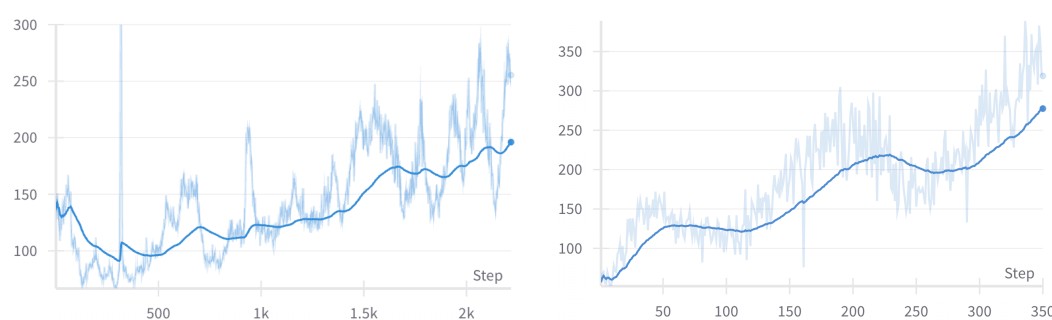

Figure 22: Response Length of JudgeLRM-3B and JudgeLRM-7B by Steps.

---

**The reasoning of JudgeLRM-7B**

`<think>` I will evaluate Assistant 1 and Assistant 2 based on the criteria of helpfulness, relevance, accuracy, and level of detail.

Assistant 1 starts with a statement about their own paleness, which is a bit whimsical and not necessary in the context. They then provide information about avoiding sunburn, which doesn't seem to answer the asked question—about tanning. They mention alternatives like a tanning bed, sunscreen, and protective clothing as ways to avoid burning. This information is generally useful and relevant. However, their response is primarily focused on preventing sunburn rather than providing information on how to tan without burning. The explanation can be more practical and customer-oriented, providing a clearer understanding to someone looking for tips on tanning.

Assistant 2 explains that there are multiple ways to get a tan while maintaining skin health. They suggest using sunscreen, which, while protective, can indeed help with tanning. They also advise to use a tanning bed carefully, which seems a fair inclusion in a variety of methods. Both assistants mention protective clothing, which is probably the most effective method to avoid sunburn and to ensure safe tanning. They both also give a brief mention of times to avoid direct sunlight for best results.

Assistant 2's response, however, provides a more comprehensive and clear explanation of different methods to get a tan without burning, while also providing context on how to protect your skin from UV rays. In addition, they explore the relation between tanning and vitamin D production, pointing out that getting a tan might be healthy for a general audience. Their response seems to be more to the point and fits the asked question a little better, also giving more contextual information that could be valuable for the reader.

Although both Assistants seem to be accurate, in the sense that the advice given does not convey any apparent misinformation, Assistant 2 provides more detailed information about individual ways to get a tan while also explaining the potential benefits and health aspects, thus more relevant to a five-year-old than Assistant 1.

In terms of helpfulness, both provide useful suggestions. Assistant 1 is a tad less clear and focused, while Assistant 2's points are slightly easier to understand and can be clearly applied to the reader's context. Assistant 2 also provides educational information about skin health and tanning, found in their reference to vitamin D benefits, which can be slightly helpful.

Assistant 1's response is more conversational, and although it provides general information, the tone seems more casual and could be confusing, while Assistant 2's is more structured and detailed, providing a clear and helpful mixture of information and advice.

`</think>`

`<answer>`6`</answer><answer>`9`</answer>`

---

Figure 23: The reasoning of JudgeLRM-7B in Case 2.

---

**Deepseek-R1's Thinking and Answer of Case 2**

Okay, so the user is asking if it's true that people can have high blood pressure without symptoms. Let's look at both assistants' answers to evaluate their performance.

Starting with Assistant 1's answer. The healthcare provider responds affirmatively, which is correct. They mention that high blood pressure can be asymptomatic, especially if not very high. That's accurate. They also note the risks like heart disease and stroke, which is good. The patient then asks how to lower blood pressure, and the provider lists diet, exercise, and stress reduction. But wait, the answer seems cut off. The formatting is a bit odd with the patient and provider dialogue. The healthcare provider's advice is correct but maybe lacks some detail. For example, they could mention reducing sodium intake, limiting alcohol, or medications if needed. Also, the response to the second patient question about lowering blood pressure is a bit brief. The answer is relevant and accurate but could be more detailed.

Now, Assistant 2's answer. Wait, the user's question ends with "And if so, how can I lower my risk of this disease? Thank you!" but Assistant 2's answer seems incomplete. Actually, looking back, the user's message might be a continuation. Wait, the user's message includes two parts: the initial question and then a follow-up. However, in the provided dialogue, Assistant 2's answer is just "And if so, how can I lower my risk of this disease? Thank you!" which seems like a repetition of the user's follow-up question, not an actual answer. That might be a formatting error. Wait, maybe the user's input was split, and Assistant 2's answer is actually the part after the user's follow-up. But looking at the structure, after the user's first question, both assistants' answers are presented. Wait, the user's message includes the question and then two assistants' answers. The first assistant's answer is a dialogue between healthcare provider and patient, addressing the initial question and follow-up. The second assistant's answer seems to be only "And if so, how can I lower my risk of this disease? Thank you!" which is not an answer but a repetition of the user's follow-up. That must be a mistake in the input formatting. Alternatively, maybe Assistant 2's answer is missing. Wait, the way it's structured: after the user's question, the user's message includes the question and then the two assistants' answers. So Assistant 1's answer is the dialogue, and Assistant 2's answer is the line starting with "And if so..." which is actually part of the user's message. Wait, no, looking at the exact structure:

The user's message is:

[Question]
I've heard that some people can have high blood pressure without having any symptoms. Is this true?
[Assistant 1's Answer]
... dialogue ...
[Assistant 2's Answer]
And if so, how can I lower my risk of this disease? Thank you!

Ah, so Assistant 2's answer is just that line, which is actually the user's follow-up question, not an answer. That's a problem. So Assistant 2's answer is incomplete or incorrectly formatted. It seems like Assistant 2's answer is just repeating the user's own follow-up question, which doesn't make sense. Therefore, Assistant 2's answer is not providing any useful information. It's either a mistake or a placeholder. So in terms of scoring, Assistant 1 provided a relevant and accurate answer, albeit a bit brief and with some formatting issues (like the dialogue structure which might be confusing). Assistant 2's answer is nonsensical as it's just repeating the user's question. Therefore, Assistant 2 gets a low score.

But wait, maybe I'm misinterpreting. Let me check again. The user's message includes the initial question and then two assistant answers. The first answer is the dialogue, and the second is that line. So Assistant 2's answer is just "And if so..." which is not an answer. So Assistant 2 failed to provide a proper response. Therefore, Assistant 1 gets points for correctness and relevance but loses some for lack of detail. Assistant 2's answer is not an answer, so it's irrelevant and unhelpful. So scores: Assistant 1 maybe a 7 (correct but could be more detailed), Assistant 2 a 1 or 2 for not answering.
</think>
<answer>7</answer><answer>2</answer>"

Figure 24: Example of Deepseek-R1's Thinking and Answer of Case 2.

**Gold Response of Case 2 from GPT-4**

9 1

Assistant 1 provided a comprehensive, accurate, and relevant answer to the user's question. The assistant explained what high blood pressure is, why some people might not have symptoms, and the risks associated with it. The assistant also provided detailed advice on how to lower blood pressure, which was very helpful. Therefore, Assistant 1 receives a score of 9.

On the other hand, Assistant 2's response was not relevant to the user's question. It seems like the assistant misunderstood the question and provided an incomplete response. The assistant did not provide any information about high blood pressure or how to lower it. Therefore, Assistant 2 receives a score of 1.

Figure 25: Gold response of Case 2 from GPT-4 (Provided in PandaLM dataset) (No reasoning from gpt-4).

