# OpenReview forum: "JudgeLRM: Large Reasoning Models as a Judge"
_ICLR.cc/2026/Conference — Submitted to ICLR 2026_

### Official Review · Reviewer_Nsgu · 2025-10-22

**Soundness:** 4
**Presentation:** 3
**Contribution:** 3
**Rating:** 8
**Confidence:** 4

**Summary:**

This work addresses one of the problems of existing LLM-as-a-judge approaches -- their poor performance on the tasks requiring complex reasoning. Authors show the negative correlation between the performance gains from SFT and proportion of reasoning-demanding samples in a given domain. To fix this gap, they propose a family of models -- JudgeLRM -- which were trained using GRPO method with "judge-wise, outcome-driven" reward function. This reward function was designed to optimize for structural correctness, relational accuracy, absolute score accuracy, and judgment confidence. Their results show that 3B JudgeLRM model is more accurate than GPT-4 on the human-annotated PandaLM benchmark, and 7B outperforms DeepSeek-R1 model. Additional analysis show that JudgeLRM gains most on the reasoning-demanding tasks were SFT models fail.

**Strengths:**

- Clear motivation: The authors show that SFT is insufficient for judging tasks requiring some extend of reasoning.
- Technical Contribution: Core contribution -- the judge-wise, outcome-driven reward function  -- is novel and give significant results.
- Strong results and deepened analysis: Authors show that JudgeLRM works better than SFT on reasoning-demanding tasks. They provide ablation studies to justify the reward-function design, and provide qualitative examples of improved reasoning.

**Weaknesses:**

- Contradictory Statistics in Analysis: In the caption of Figure 3, the authors mention the negative linear trend, but both the plot and the equation parameters (y = 0.2x − 1.05) seem positive. In the Section 4.3 the authors also mention "we observe a correlation coefficient of 0.20 between relative improvement and reasoning rate", which imply $R^2=0.04$. In the Figure 3 caption the $R^2=0.95$ is given.
- "Reasoning" Labeling Methodology: In the two most important figures (Figure 1 and Figure 3) the authors use "Proportion Requiring Reasoning to Judge (%)" metric. The process of creating this metric is only revealed in the appendix. While authors did validate this method against human annotators it still creates a potential circularity. The paper essentially uses GPT-4's own definition of "reasoning" to motivate the need for a new model. The paper then uses this new model to claim it has surpassed the performance of GPT-4 itself. Given that this metric is so central to the paper's argument, the use of an LLM to generate it should be discussed transparently in the main paper, not just in the appendix.
- Overstated claims about surpassing GPT-4: JudgeLM dataset used to train JudgeLRM was made entirely from GPT-4 answers. The reward function was design to match the final outcomes from GPT-4 labels. Taking that into consideration, claiming that JudgeLRM "surpasses GPT-4" may be too strong. More precise framing, that this specialized 3B model was able to outperform its general-purpose teacher (GPT-4) on a different test (the human-annotated PandaLM benchmark). This is still a very impressive result, but this description is more accurate.

**Questions:**

I don't have specific questions, but I would be more than happy to see improvements stated in the weaknesses section.

---

> ### Author Response · Authors · 2025-11-21
>
> We appreciate the reviewer's scrutiny regarding the reliability of GPT-4 labels and the framing of our performance claims. We have conducted additional verification and refined our claims as follows:
>
> 1. The validity of "Reasoning Requirement" categorization:
>
> We appreciate the opportunity to clarify the robustness of our analysis. A core finding of our work (Fig. 1 & 3) is that standard SFT fails in **reasoning-intensive domains**, while JudgeLRM significantly improves in them. To ensure this insight is not an artifact of arbitrary task classification, or circularity, we validated the "Reasoning Requirement" labels (the 1-10 difficulty scores assigned by GPT-4) using GPT-5.1-2025-11-13. Re-evaluating the PandaLM test set revealed a high degree of consensus on reasoning difficulty between the teacher (GPT-4) and the advanced evaluator (GPT-5.1): Exact Match (Difference = 0): 34.1%, Adjacent Match (Difference = 1): 41.3%. Over 75% of tasks received a difficulty rating within 1 point of the GPT-4 label. The Spearman correlation (ranking relatively) of GPT-4 and GPT-5.1 labels is $\rho=0.4515$, $p < 1e^{-24}$**.** While the vast majority of the reasoning requirement scores are clustered in the PandaLM test set, rank-based metrics like Spearman are disproportionately sensitive to minor variances. The over 75% adjacent agreement confirms that the GPT-4 reasoning requirement labeling is stable and objective, and the trends observed in Figure 1 and Figure 3 are robust. The fact that JudgeLRM outperforms SFT specifically on these reliably identified reasoning-intensive tasks (validated by GPT-5.1) provides strong evidence that our RL method successfully activates reasoning capabilities, rather than simply fitting dataset noise.
>
> 2. Generalization via Reasoning beyond Imitation:
>
> We acknowledge that training on GPT-4 labels carries a risk of overfitting. However, our key contribution is using RL to instill a structured **reasoning process**. This reasoning capability enables the model to learn *how to judge* rather than merely mimicking the teacher's score distribution. The compelling proof is JudgeLRM's performance on the human-annotated and out-of-distribution PandaLM benchmark. As shown in Table 3, JudgeLRM (7B) achieves an F1 score of 75.05, explicitly outperforming its teacher GPT-4. This confirms that JudgeLRM has acquired **generalized reasoning capabilities** that align better with human judgments than the original teacher model, rather than **overfitting** to the features of the teacher model.
>
> (Furthermore, our train set is GPT-4-labeled JudgeLM, which the reliability of GPT-4 as a reliable judge has been rigorously validated in the original JudgeLM study and MT-Bench [1] . Their experiments demonstrated that GPT-4 achieved higher agreement with the human ground truth (84.48%) than a second human annotator did (79.82%). The authors attributed this to GPT-4's superior adherence to detailed annotation instructions regarding relevance and detail.)
>
> 3. Refining Claims and Typos
> We agree with the reviewer’s suggestion to be precise. We revised our claim to state that our **"**on the human-annotated out-of-distribution PandaLM benchmark, JudgeLRM-3B/4B surpasses its general-purpose teacher GPT-4, with particularly strong gains on reasoning-heavy tasks.**"** .
>
> We apologize for the typo in the caption of Figure 3. We have corrected it to explicitly describe a **positive** linear trend, which aligns with the plotted data, the equation ($y=0.2x - 1.05$), and our analysis in Section 4.3.
>
> We have revised the Introduction to explicitly disclose that the 'reasoning-required' samples are labeled by GPT-4, ensuring immediate transparency.
>
> [1] Judging LLM-as-a-Judge with MT-Bench and Chatbot Arena. NIPS 2023 DB Track.

---

> > ### Comment · Reviewer_Nsgu · 2025-11-25
> >
> > Thank you for your detailed responses, the additional experiments, and the revisions to the manuscript. I am maintaining my score of 8, as I believe this work is of sufficient quality for acceptance and presentation as a conference poster.

---

> > > ### Author Response · Authors · 2025-12-01
> > >
> > > We are thrilled to receive your positive assessment and strong support! We deeply appreciate your recognition of the quality and contribution of our work, and we look forward to the opportunity to present it to the community.

---

### Official Review · Reviewer_g7Nk · 2025-10-25

**Soundness:** 2
**Presentation:** 1
**Contribution:** 2
**Rating:** 2
**Confidence:** 4

**Summary:**

This paper introduces JudgeLRM, a family of reinforcement learning–trained models designed for judgment tasks that demand complex reasoning. The central hypothesis is that many judgment and reward-scoring tasks inherently require reasoning, which limits the effectiveness of standard SFT-based judges and reward models. To address this, the authors propose an RL framework that trains models to reason and evaluate by rewarding them based on the accuracy of their predicted scores relative to gold human or frontier model judgments. Experiments demonstrate that JudgeLRM models outperform both SFT and baselines such as BT and DPO, highlighting the importance of using reasoning and RL for developing better judges.

**Strengths:**

* The motivation is clear: judgment tasks are inherently reasoning-intensive, and this work provides an interesting attempt to align reward learning with reasoning ability.
* The empirical results are strong, showing consistent improvements over SFT and existing baselines (e.g., BT, DPO).

**Weaknesses:**

* **Restricted applicability**: The framework seems limited to settings where ground truth scores are available. It’s unclear how it extends to preference-based or binary feedback data.
* **Writing**: The paper is poorly written. The method is not introduced properly, the introduction is too experimental, the results are not written cleanly. The baselines and experimental design can be explained significantly better.
* **Heuristic reward design**: The absolute reward formulation and its hyperparameters appear heuristic and tuned to specific score distributions, raising concerns about generalizability.
* **Confidence reward is poorly motivated**: It’s unclear why a non-continuous confidence reward is used, and it seems to specifically encourage overconfidence rather than calibrated predictions.
* **Pairwise-only framework**: Since the model is trained pairwise, it may struggle to produce meaningful single-response scores, limiting its inference-time utility.
* **Weak case study**: The case study relies purely on qualitative analysis, without quantitative evidence to substantiate claims about emergent complex reasoning behaviors (e.g., verification. subgoal setting).
* **Limited related work**: The related work section is underdeveloped. There have been a few other works training reasoning reward models. Even if they are concurrent, some more discussion is necessary.
* **Poor presentation choices**: Table 1 placement is suboptimal—this section should feature a main figure summarizing the method or key results.
* **Cluttered introduction**: The introduction includes excessive experimental discussion. These details would fit better in the Motivation or Background section.

**Questions:**

1. How would the framework handle preference data (e.g., pairwise comparisons without numeric scores)?
2. Can you clarify the motivation and formulation of the confidence reward? Why is it not continuous or calibrating?
3. I am highly doubtful about the generalizability of the method to different score distributions. For example if the ground truth scores are distributed between 0-100, then reward hyperparameters need to be tuned accordingly.
4. The claim that these rewards encourage calibrated confidence is probably incorrect as no proper scoring rule [2] was used in the reward function. Can the authors argue why this should be true?
5. How were the reward hyperparameters chosen, and how sensitive are results to these values?
6. Since the framework is pairwise by design, how is it adapted for single-response evaluation, and how reliable is that setup? If single-response evaluation is possible, it would be useful to have a baseline trained specifically for that setting to test if relational reasoning is truly necessary.
7. Could you provide explicit equations for the baselines, especially DPO? Is the DPO implementation just the standard version? It is okay to put these into the appendix if space is a constraint.
8. Could you compare against pairwise preference models trained explicitly for this setting (e.g., [1] but also used in multiple other works)?
9. In the “length reward” experiment, why was a threshold-based reward (120 tokens) chosen instead of a more standard continuous reward formulation? Why was 120 tokens picked as a threshold?
10. Can the authors formally define the training setting and data assumptions—specifically, the use of continuous scores between [0,10]—and discuss how this constrains generalizability?


[1]: Munos, R., Valko, M., Calandriello, D., Azar, M. G., Rowland, M., Guo, Z. D., ... & Piot, B. (2024, July). Nash learning from human feedback. In Forty-first International Conference on Machine Learning.

[2]: Gneiting, T., & Raftery, A. E. (2007). Strictly proper scoring rules, prediction, and estimation. Journal of the American statistical Association, 102(477), 359-378.

---

> ### Author Response · Authors · 2025-11-21
>
> **About applicability**:
>
> We would like to clarify that our framework is naturally compatible with preference-based or binary feedback via the **Relation Reward ($\mathcal{R}_{rel}$)** component, which optimizes for relative ranking (win/loss/tie) rather than specific scalar values.
>
> We empirically validated this in our **Ablation Study (Table 4)**. The variant trained **only with the relation reward** (denoted as w/o. R_abs + R_conf) effectively operates in a pure preference-based setting without ground truth scalar supervision. This preference-only variant achieved an **F1 score of 70.36** on the PandaLM benchmark. Remarkably, even without scalar ground truth, this performance still significantly surpasses GPT-4.
>
> This demonstrates that JudgeLRM can be effectively extended to scenarios with only binary or preference feedback, maintaining superior performance over general-purpose models by leveraging the reasoning structure driven by relational signals alone.
>
> **About writing**:
>
> We appreciate the constructive feedback on the manuscript's structure. We are actively revising the paper to replace Table 1 and expand the related work. We will upload a revised PDF containing these improvements early in the discussion period for your review.
>
> **On the heuristic reward design/generalization/calibration:**
>
> We acknowledge that the hyperparameters for the absolute reward are empirically tuned. However, we argue that this design does not hinder generalization. Instead, it serves to align the model's internal quality assessment with a standardized output scale, while the core learning objective is the **reasoning capability** itself.
>
> The absolute reward ($\mathcal{R}_{absolute}$) acts as a boundary condition to teach the model how to map its internal quality analysis to a specific 1-10 scale. The RL process focuses on optimizing the **reasoning path** (via the structural and outcome rewards) to accurately identify quality differences. Once the model learns to "analyze" quality effectively, mapping this analysis to a score is a generalizable skill that transcends specific domain distributions. The model learns the *logic of judgment* rather than memorizing the *distribution of scores*.
> Prior research [1] supports the validity of the LLM scoring approach. [1] demonstrated that LLMs (specifically GPT-4, our teacher model) possess a **"relatively stable internal rubric"** for single-answer grading. Their findings show that point-wise grading achieves high agreement with human preferences (over 80%) across diverse domains (e.g., Humanities, STEM, Coding), verifying that the ability to map quality analysis to a scalar score is robust and domain-agnostic. Our method distills this stable internal rubric into JudgeLRM.
>
> The strongest evidence against overfitting is our performance on the human-annotated out-of-domain PandaLM benchmark, with a different distribution than the GPT-4-labeled JudgeLM training set. Despite the heuristic tuning on JudgeLM, JudgeLRM-7B achieves an F1 score of 75.05 on PandaLM, significantly outperforming GPT-4 (61.8). This confirms that the absolute reward successfully guided the model to acquire generalized reasoning capabilities that remain effective in strictly human-aligned, out-of-distribution scenarios.
>
>  [1] Judging LLM-as-a-Judge with MT-Bench and Chatbot Arena. NIPS 2023 DB Track.

---

> > ### Author Response · Authors · 2025-11-21
> >
> > **About confidence reward**:
> >
> > We clarify that the confidence reward ($\mathcal{R}_{conf}$) is explicitly designed to address the **"indecision" problem** in hard samples, where models tend to assign dangerously similar scores to different quality responses. We chose a **non-continuous (discrete)** design specifically to prevent the uncalibrated overconfidence that the reviewer is concerned about.
> >
> > To demonstrate this, we conducted an additional ablation study comparing our method against a **continuous confidence reward** variant (defined as $r = 0.1 \times |s_1 - s_2|$) and a version removing it entirely.
> >
> > **Continuous Reward Causes Extreme Overconfidence and Collapse:** As hypothesized by the reviewer, a continuous reward incentivizes the model to maximize the score margin infinitely. As shown in the table below, the continuous variant collapsed to an F1 of 56.12. We observed that this model learned to "game" the reward system by outputting extreme scores (e.g., **0 vs 10**) regardless of the actual quality difference, leading to severe uncalibration.
> >
> > **Discrete Reward Balances Decisiveness and Calibration:** Our discrete approach acts as a bounded bonus rather than an infinite gradient. It encourages the model to be "decisive" (make a clear judgment) only when the ranking is correct, without incentivizing extreme polarization. The discrete reward improves performance from 72.36 (w/o $\mathcal{R}_{conf}$) to 75.05 in the first version ablation study, successfully solving the tie-breaking difficulty in hard samples while maintaining a grounded score scale.
> >
> > | Metric | Agreement | Precision | Recall | F1 |
> > | --- | --- | --- | --- | --- |
> > | w/o $\mathcal{R}_{conf}$ | 77.08 | 71.36 | 75.14 | 72.36 |
> > | w. Continuous $r_{\text{conf}}=0.1 \times \Delta$ | 74.37 | 62.52 | 57.19 | 56.12 |
> > | **JudgeLRM-7B (Discrete)** | **78.28** | **74.90** | **75.74** | **75.05** |
> >
> > **About pairwise-only framework**:
> >
> > We respectfully clarify that our framework is not limited to pairwise settings. We explicitly evaluated JudgeLRM in a single-response setting (using the prompt in Figure 4), and the results are reported in Table 3 (rows marked "JudgeLRM-single").
> >
> > As shown in Table 3, even when evaluating responses individually without a reference pair, JudgeLRM-8B-single achieves an F1 score of 65.30, which still outperforms the teacher model GPT-4. This confirms that the reasoning capabilities acquired via RL are **transferable**: the model learns the abstract logic of mapping analysis to quality, rather than merely overfitting to the pairwise format.
> >
> > **Failure of SFT in Generalization (New Analysis):** To further illustrate JudgeLRM's superiority over SFT in this regard, we tested the SFT baseline (Qwen3-8B-Instruct-SFT) using the same single-response prompt.
> >
> >  Despite being explicitly prompted to provide a *single* score for *one* response, the SFT model suffered from severe format hallucination due to rote memorization. For instance, in Case ID 549, the SFT model output: *"\<think\>Assistant provided a detailed list of key points about the history of the USA, including significant events such as the founding of the United States, the signing of the Declaration of Independence and the Constitution, the Civil War, and the ratification of various amendments. However, the answer was a bit repetitive, which is why it didn't receive a perfect score.\</think\>The pair-wise score of two responses are 8 10."* even though only the first response was provided.
> >
> > In contrast, JudgeLRM correctly adapted to the single-response instruction, generating a coherent analysis followed by a single score. This contrast proves that while SFT models memorize the "pairwise output pattern," JudgeLRM learns the **generalized ability to reason**, allowing it to function effectively in both pairwise and pointwise inference settings.

---

> ### Author Response · Authors · 2025-11-21
>
> **About emergent complex reasoning behaviors**:
>
> To rigorously quantify the emergent reasoning behaviors (e.g., verification, subgoal setting), we conducted a systematic quantitative analysis on the test results using GPT-5.1-2025-11-13 as an impartial evaluator.
>
> We detailed the analysis of mapping those micro-behaviors to macro-abilities (deduction, induction, abduction in [2,3]) in Appendix Evaluation of Reasoning Capabilites in JudgeLRM, quantifying specific micro-behaviors (like *Verification*) in isolation can be noisy. Instead, we aggregated these behaviors into three fundamental **Macro-Level Meta-Abilities** that serve as their logical foundation:
>
> - **Deduction (e.g. Verification):** Applying rules to check facts.
> - **Induction (e.g. Subgoal Setting):** Synthesizing signals into a score.
> - **Abduction (e.g. Decision Justification):** Formulating the best explanation for a verdict, especially in ambiguous cases.
>
> The frequency analysis (Table below) reveals a fundamental shift in cognitive processing driven by RL:
>
> | Model (%) | Deduction (e.g., Verification) | Induction (e.g., Synthesis) | **Abduction (e.g., Justification)** |
> | --- | --- | --- | --- |
> | SFT-3B | 22.4 | 68.1 | 22.2 |
> | **JudgeLRM-3B** | **42.6** | **68.6** | **75.8** |
>
> The most striking result is the **3.4x increase in Abduction** (22.2 to 75.8). While SFT models primarily rely on *Induction* (pattern matching), JudgeLRM has learned to perform *Abduction*—the complex reasoning required to construct a compelling justification for why one response is superior. This explicitly quantifies the emergence of **Decision Justification** capabilities. Moreover, the doubling of **Deduction** counts (22.4 to 42.6) confirms that JudgeLRM is significantly more active in **Verification** and error identification compared to the SFT baseline. These quantitative metrics confirm that the complex behaviors observed in our case studies are not isolated anecdotes but statistically significant emergent properties of the JudgeLRM training process.
>
> **About hyperparams**:
>
> For the reinforcement learning algorithm, we adopted the GRPO framework. To ensure training stability and reproducibility, we strictly followed the implementation and hyperparameter settings (e.g., group size $G=8$, clip range $\epsilon=0.5$, KL coefficient $\beta=0.001$) established in Logic-RL. Other standard training parameters (e.g., learning rates, batch sizes) were selected based on pilot runs and are detailed in Section 4.1 "Implementation Details".
>
> We also explicitly analyzed the sensitivity of the reward scalars in Appendix Further Reward Design Analysis. Our experiments yielded two key conclusions:
>
> - **Robustness to Exact Values:** The method is largely **insensitive** to the precise numerical values of the rewards. For instance, slightly modifying the reward scalars (e.g., Relation Reward $\pm 0.1$) resulted in statistically insignificant performance variations.
> - **Sensitivity to Relative Hierarchy:** However, the **relative order** of the reward components is critical. As shown in Table 10, altering the reward hierarchy (e.g., making the absolute score reward dominant over the relational reward) caused a significant drop in performance (F1 decreased from 72.12 to 69.31). This confirms that prioritizing relational reasoning over absolute scoring is essential for the model's success.
>
> **About the baseline equations**:
>
> We have provided the explicit mathematical formulations for all baselines in Appendix Formalization of Baseline and JudgeLRM Objectives. We also added the experiments on distilling Deepseek-R1 reasoning chain in SFT-Distill-R1-Think (Qwen2.5-3B-Instruct) in Table 2.
>
> As detailed in the Appendix, our JudgeLRM framework distinguishes itself from DPO and other baselines through three fundamental mechanisms:
>
> **Explicit Policy Gradient:** Unlike the RL-free nature of DPO, JudgeLRM utilizes explicit policy gradients with a normalized advantage function.
>
> **Group-wise Normalization:** We leverage Group-wise Normalization to enhance training stability and generalization, particularly across heterogeneous task domains, a feature absent in both DPO and standard PPO.
>
> **Structure-Aware Reward:** Our method employs a combined reward ($r_i$) that explicitly enforces structural adherence ($R_{struct}$) alongside content accuracy. This directly trains the model to produce robust, structured reasoning paths ($\langle think \rangle$) alongside the final scores, which standard preference optimization cannot easily enforce.
>
> [2] Cognitive behaviors that enable self-improving reasoners, or, four habits of highly effective stars. COLM 2025.
>
> [3] Beyond 'Aha!': Toward Systematic Meta-Abilities Alignment in Large Reasoning Models. https://arxiv.org/abs/2505.10554

---

> > ### Author Response · Authors · 2025-11-21
> >
> > **About pairwise preference models**:
> >
> > We clarify that our objective differs fundamentally from methods like NLHF. NLHF typically aims to train a Policy/Generator to align with human preferences (i.e., "how to generate better text"). In contrast, JudgeLRM aims to train a Judge/Evaluator (i.e., "how to accurately evaluate which text is better").
> >
> > These approaches are complementary, not competing. JudgeLRM serves as the high-quality verification mechanism (metric) required to evaluate or train generative models (like those in NLHF/RLHF).
> >
> > Regarding the comparison with "pairwise preference models" trained for this setting, we did evaluate against the standard representatives of this paradigm in Table 2:
> >
> > **Bradley-Terry Model**: A purely pairwise preference model that predicts a scalar reward to derive win/loss probabilities.
> >
> > **CLS-RM**: A classification-based reward model trained explicitly on preference pairs.
> >
> > As formalized in Appendix Formalization of Baseline and JudgeLRM Objectives, the key difference is that standard pairwise models (like BT or DPO) abstract judgment into a single latent probability ($P(A_1 > A_2)$). In contrast, JudgeLRM explicitly models the reasoning process ($\langle think \rangle$).
> >
> > As shown in Table 2, JudgeLRM-3B (F1 72.12) significantly outperforms the Bradley-Terry baseline (F1 58.94). This confirms that for the judging task, merely modeling pairwise probability is insufficient; the model must learn the logic of evaluation through reasoning.
> >
> > **About length reward**:
> >
> > We clarify that the "Length Reward" experiment was designed as a **negative control** to investigate whether our performance gains stemmed simply from longer outputs (verbosity) or genuine reasoning quality.
> >
> > We deliberately chose a **threshold-based (step function)** reward over a continuous one to prevent **reward hacking**. A continuous reward (e.g., $r \propto \text{length}$) incentivizes the model to generate infinite, irrelevant text to maximize the score. The threshold design ($L=120$) acts as a "minimum sufficiency" constraint, encouraging the model to provide a rationale of adequate depth without incentivizing unbounded verbosity.
> > The threshold of 120 tokens was empirically selected. To address the reviewer’s question regarding sensitivity, we conducted an additional experiment increasing the threshold to 300 tokens.
> >
> > | Method | Agreement | Precision | Recall | F1 |
> > | --- | --- | --- | --- | --- |
> > | **JudgeLRM-7B (Ours)** | **78.28** | **74.90** | **75.74** | **75.05** |
> > | w. $\mathcal{R}_{\text{length}}$ ($\mathbf{L}=120$) | 78.28 | 75.81 | 69.19 | 71.34 |
> > | w. $\mathcal{R}_{\text{length}}$ ($\mathbf{L}=300$) | 76.98 | 73.80 | 70.68 | 71.98 |
> >
> > As shown above, explicitly incentivizing length, whether at 120 or 300 tokens, consistently **degrades performance** compared to our proposed method (F1 75.05). This reinforces our conclusion in Section 4.4 that JudgeLRM's superiority is driven by **structured reasoning quality** (enforced by $\mathcal{R}*{struct}$ and $\mathcal{R}*{content}$), not by the mere quantity of tokens generated.

---

> > > ### Comment · Reviewer_g7Nk · 2025-11-24
> > > **Response to rebuttal**
> > >
> > > I thank the authors for the detailed rebuttal. The empirical results are strong, as I mentioned in my original review. However, I will maintain my scores due to several key concerns:
> > >
> > > 1. **Extremely Heuristic Reward Design**: The reward design is not principled at all and can lead to severe miscalibration in many settings. I had cited a paper on scoring rules in my original review. The authors tried a continuous confidence reward $|s_1 - s_2|$, however this is not a proper scoring rule. Optimizing this reward will lead to extreme predictions because that is the optimal thing to do! I have the same concern with the “confidence reward” used by the authors in the original paper. Giving an arbitrary reward of 0.2 if the absolute difference of predicted scores is greater than the absolute difference of ground truth scores is not a principled design decision. It might work well in the settings the authors consider, but can cause issues in other settings. For example, if two responses $y_1$ and $y_2$ are almost identical (very difficult to separate), then the confidence reward puts optimization pressure to forcefully separate them. In other words, the bounded bonus of the confidence reward can affect the training in undesirable ways. I think it is possible to design a principled, well-motivated reward function for this setting which removes the need for having separate relation, absolute and confidence rewards. Having three separate rewards (combined with the need to tune them) might lead to some empirical gains, but I am not convinced they are necessary or even needed.
> > > 2. **Writing**: The writing of the paper is poor, it can be much better-motivated. I think it is currently under the standard required for ICLR.
> > > 3. **Related Work**: First, I had cited NLHF. When comparing against Judge-LRM, I was referring to the pairwise preference model which is trained in NLHF. This preference model is then used to train a policy (which is not relevant here). I was asking authors to compare against this pairwise preference model setup. More concretely, training a pairwise BT model which takes $x,y_1,y_2$ as a single input. There are also other (possibly) concurrent works that are using RL to train judges [1]. Even if they are concurrent, some discussion of them would be nice, as mentioned in my original review.
> > >
> > > I will also track the responses from other reviewers. In case I feel there is a strong case for raising the score, I will consider it.
> > >
> > > [1]: Whitehouse, C., Wang, T., Yu, P., Li, X., Weston, J., Kulikov, I., & Saha, S. (2025). J1: Incentivizing thinking in llm-as-a-judge via reinforcement learning. arXiv preprint arXiv:2505.10320.

---

> > > > ### Author Response · Authors · 2025-12-01
> > > >
> > > > 1. On “Extreme Heuristic Reward” and “Lack of Theoretical Basis”
> > > >
> > > > We understand the reviewer’s concern that our confidence reward might be an “extreme heuristic” lacking theoretical grounding.  If one directly uses a linear, unbounded continuous reward proportional to the margin (e.g., $R \propto |s_1 - s_2|$), then pushing the scores to be as far apart as possible is indeed the optimal behavior, but it would cause problems both theoretically and practically.
> > > >
> > > > However, our actual reward design differs fundamentally from the reviewers' concerns:
> > > >
> > > > - Our reward is piecewise, bounded, and hierarchically dependent;
> > > > - The confidence reward is only a minor component and is strongly constrained by the priority of relation rewards;
> > > > - Within the GRPO optimization framework, it is dynamically equivalent to a hinge loss-like margin-satisficing objective,  rather than an unbounded “more is always better” scoring rule.
> > > >
> > > > As a result, the failure mode the reviewer correctly identifies for linear continuous rewards does not arise under our specific reward structure.
> > > >
> > > > 2. The Empirical Evidence on ablation: Continuous and Discrete/Bounded Confidence Rewards
> > > >
> > > > In our previous response, we already added ablation studies to empirically examine exactly the kind of continuous confidence reward the reviewer is worried about.
> > > >
> > > > | **Configuration** | **Agreement** | **Precision** | **Recall** | **F1** |
> > > > | --- | --- | --- | --- | --- |
> > > > | **JudgeLRM-7B (Ours)** | **78.28** | **74.90** | **75.74** | **75.05** |
> > > > | - w. continuous r_conf | 74.37 | 62.52 | 57.19 | 56.12 |
> > > >
> > > > As shown in the table, replacing our discrete reward with a continuous linear variant ($r_{\text{conf}} \propto \Delta$) leads to a drastic performance drop (F1: 75.05 $\to$ 56.12), which only judgs the answer score 1 or 10.  It suggests that our discrete, bounded reward structure is critical for preventing model extremization.
> > > >
> > > > 3. On “Forcing Ties Apart”: Hierarchical Dependency Design
> > > >
> > > > We claim that a confidence reward **will not** pressure the model to “break ties” when the ground-truth label is a tie. We introduced a strict **hierarchical dependency** in the reward. The Relation Reward ($R_{\text{rel}}$) dominates in magnitude and acts as a prerequisite: the smaller Confidence Reward ($R_{\text{conf}}$) activates only when the predicted relation is correct (i.e., $\text{sign}(s_1 - s_2) = \text{sign}(s_1^* - s_2^*)$). Consequently, for true ties, 'forcing a separation' fails the relation check and forfeits the dominant $R_{\text{rel}}$, ensuring that correctly predicting a tie yields the maximum total reward.
> > > >
> > > > 4.  On “Barely Distinguishable but Non-Tie” Pairs
> > > >
> > > > It is crucial to distinguish 'true ties' from 'subtle differences.’ Our design targets the 'central-tendency bias' prevalent in existing judges, where **subtle differences are often ignored,** thus collapsing predictions to ties. While the hierarchical structure protects true ties, the confidence reward encourages the model to capture fine-grained advantages in non-tie pairs. Importantly, this mechanism **prevents overconfidence through saturation**: the reward stops increasing once the predicted margin matches the ground truth or a fixed threshold. This ensures the model learns to discriminate sufficiently without exaggerating subtle differences.
> > > >
> > > > 5. The Theoretical Perspective: Discrete Confidence Reward and Hinge Loss-Like Dynamics
> > > >
> > > > From a theoretical standpoint, we explicitly avoid linear margin rewards to prevent extremization. Let $\Delta = |s_1 - s_2|$ be the predicted margin and $\Delta^*$ be the target.
> > > >
> > > > A continuous linear reward $R_{\text{cont}} = \alpha \Delta$ implies a constant positive gradient $\frac{dR}{d\Delta} = \alpha > 0$ for all $\Delta$. This provides an unbounded incentive, pushing the model to *maximize the margin indefinitely* regardless of the ground truth.
> > > >
> > > > In contrast, our discrete confidence reward functions as a bounded, margin-satisficing objective:
> > > >
> > > > $$R_{\text{disc}}(\Delta) = \beta \cdot \mathbb{I}[\Delta \ge \Delta^*]$$
> > > >
> > > > Under RL optimization, this **aligns with the dynamics of a reverse Hinge Loss** ($L_{\text{hinge}} = \max(0, \Delta^* - \Delta)$). The crucial property lies in the gradient saturation:
> > > >
> > > > When $\Delta < \Delta^*$: The model receives a signal to increase the margin.
> > > >
> > > > When $\Delta \ge \Delta^*$: The reward saturates, and the effective gradient vanishes ($\frac{dR}{d\Delta} \to 0$).
> > > >
> > > > This (margin-based learning) saturation mechanism ensures the model is incentivized only to **satisfy** the target margin, not to **exaggerate** it, effectively immunizing the model against the extremization risks inherent in linear rewards.

---

> > > > > ### Author Response · Authors · 2025-12-01
> > > > >
> > > > > 6. “Hierarchical + Bounded” aligns with GRPO design
> > > > >
> > > > > This decomposition is essential for the stability of GRPO, which estimates advantage via group normalization: $A_i = \frac{r_i - \mu}{\sigma}$.
> > > > >
> > > > > **Boundedness ensures stability:** GRPO normalizes rewards within a group. Unbounded rewards introduce extreme outliers that skew the group statistics ($A_i = \frac{r - \mu}{\sigma}$), rendering gradients unstable and variance explotion. Our bounded, hinge-like reward ensures that all correct samples yield comparable scores, keeping variance controlled and focusing optimization on improving incorrect samples rather than inflating margins endlessly.
> > > > >
> > > > > **Hierarchy enforces prioritization:** The hierarchical magnitude ($R_{\text{rel}} \gg R_{\text{conf}}$) forces the model to prioritize relation correctness before optimizing confidence. This effectively shapes the loss landscape to penalize 'confidently wrong' predictions heavily, a safeguard that flat linear rewards lack.
> > > > >
> > > > > 7. On the Necessity of Three Reward Components
> > > > >
> > > > > While a unified reward is theoretically possible, our decomposed design addresses distinct objectives required for a robust judge: **Correctness ($R_{\text{rel}}$)**, **Global Scale ($R_{\text{abs}}$)**, and **Granularity ($R_{\text{conf}}$)**. Rather than relying on a single complex heuristic, this structure provides:
> > > > >
> > > > > - Anchoring: $R_{\text{abs}}$ prevents the 'floating scale' problem common in pairwise comparisons.
> > > > > - Stability: Independent bounding of components prevents any single objective from destabilizing the GRPO updates.
> > > > > - Performance: Empirical results confirm that this modular design outperforms simpler variants, balancing the need for strict ordering with the ability to distinguish subtle margins.
> > > > >
> > > > > To summarize point by point:
> > > > >
> > > > > (1) We agree that a linear continuous confidence reward $R \propto |s_1 - s_2|$ drives extremization and miscalibration; our PandaLM ablations confirm this, and we explicitly avoid such a design.
> > > > >
> > > > > (2) Our confidence reward is not an unprincipled heuristic but behaves like a bounded, hinge-style objective that rewards margins up to a target and then saturates, preventing unbounded incentives to be extreme.
> > > > >
> > > > > (3) Because the relation reward is much larger and confidence is only active when the relation is correct, the model is not incentivized to break true ties; doing so is strictly suboptimal under our reward.
> > > > >
> > > > > (4) For nearly indistinguishable but non-tie pairs, our design intentionally encourages the model to overcome central-tendency bias and make modest, bounded distinctions aligned with ground truth, rather than exaggerating differences.
> > > > >
> > > > > (5) The three-component decomposition, together with hierarchical activation and boundedness, is well aligned with the mechanics of GRPO: it stabilizes group-normalized advantages, suppresses “confidently wrong” solutions, and separates correctness, scale, and margin calibration into interpretable, complementary objectives.

---

> > > > > > ### Author Response · Authors · 2025-12-01
> > > > > >
> > > > > > About Writing:
> > > > > >
> > > > > > We are confident that the revised version addresses your concerns regarding the writing standards. We believe the presentation quality matches the empirical strength of the work. As noted by Reviewer Nsgu, the paper possesses **"strong results and deepened analysis"** and is **"of sufficient quality for acceptance."** If there are specific sections where the logic remains unclear, we would be very grateful if you could point them out. We remain committed to perfecting the manuscript.
> > > > > >
> > > > > > About Related Work:
> > > > > >
> > > > > > In our revised version, we have significantly strengthened the Related Work section to address your insightful suggestions:
> > > > > >
> > > > > > - **Architecture Comparison:** We explicitly positioned NLHF [1] (and RLHF) as discriminative approaches optimized for **preference signal aggregation**. We contrasted these "System 1" scalar predictors with JudgeLRM's "System 2" generative paradigm, highlighting the fundamental shift from implicit scoring to explicit, verifiable reasoning.
> > > > > > - **Differentiation from Concurrent Works:** We integrated discussions on concurrent System 2 research (e.g., J1 [2], RM-R1[3]). We clarified JudgeLRM's distinct ecological niche: focusing on **efficient emergence** in compact models using solely outcome-driven rewards, distinguishing our work from approaches that rely on extensive process supervision or massive inference-time scaling.
> > > > > > - **Theoretical Depth:** We added a discussion characterizing the judge task as **abductive/evaluative reasoning**, differentiating its open-ended cognitive mechanism from the deductive logic typical of math/code tasks.
> > > > > >
> > > > > > [1] Nash Learning from Human Feedback. ICML 2024.
> > > > > >
> > > > > > [2] J1: Incentivizing Thinking in LLM-as-a-Judge via Reinforcement Learning. https://arxiv.org/abs/2505.10320.
> > > > > >
> > > > > > [3] RM-R1: Reward Modeling as Reasoning. https://arxiv.org/abs/2505.02387.

---

> > > > > > > ### Author Response · Authors · 2025-12-01
> > > > > > >
> > > > > > > About training a pairwise BT model which takes $x,y_1,y_2$ as a single input:
> > > > > > >
> > > > > > > We implemented the Single-Input Pairwise Bradley-Terry (BT) Model exactly as requested. We trained a Cross-Encoder (based on Qwen2.5-3B) that takes $(x, y_1, y_2)$ as input and optimizes a Binary Cross-Entropy loss using soft labels derived from the score difference $\sigma(k \cdot (s_1 - s_2))$, aligning with the standard BT formulation.
> > > > > > >
> > > > > > > ```python
> > > > > > > PAIR_TEMPLATE = (
> > > > > > >     "<|im_start|>system\n"
> > > > > > >     "You are a strict evaluator. Compare two candidate answers and decide "
> > > > > > >     "which one better addresses the user.\n"
> > > > > > >     "<|im_end|>\n"
> > > > > > >     "<|im_start|>user\n{question}<|im_end|>\n"
> > > > > > >     "<|im_start|>assistant\n"
> > > > > > >     "Response A:\n{answer_a}\n\n"
> > > > > > >     "Response B:\n{answer_b}\n\n"
> > > > > > >     "Which response is better? Answer with only 'A' or 'B'.\n"
> > > > > > >     "<|im_end|>"
> > > > > > > )
> > > > > > >
> > > > > > > class CrossBTHead(nn.Module):
> > > > > > >     def __init__(self, hidden_size: int):
> > > > > > >         super().__init__()
> > > > > > >         self.proj = nn.Linear(hidden_size, 1)
> > > > > > >
> > > > > > >     def forward(self, hidden_state: torch.Tensor) -> torch.Tensor:
> > > > > > >         return self.proj(hidden_state).squeeze(-1)
> > > > > > >
> > > > > > >
> > > > > > > class CrossBTModel(nn.Module):
> > > > > > >     def __init__(self, base: AutoModel):
> > > > > > >         super().__init__()
> > > > > > >         self.base = base
> > > > > > >         self.head = CrossBTHead(base.config.hidden_size)
> > > > > > >
> > > > > > >     def encode(self, input_ids: torch.Tensor, attention_mask: torch.Tensor) -> torch.Tensor:
> > > > > > >         outputs = self.base(input_ids=input_ids, attention_mask=attention_mask)
> > > > > > >         hidden = outputs.last_hidden_state
> > > > > > >         seq_lens = attention_mask.sum(dim=1) - 1
> > > > > > >         batch_indices = torch.arange(hidden.size(0), device=hidden.device)
> > > > > > >         eos_states = hidden[batch_indices, seq_lens]
> > > > > > >         return eos_states
> > > > > > >
> > > > > > >     def forward(self, batch: Dict[str, torch.Tensor]) -> Dict[str, torch.Tensor]:
> > > > > > >         pooled = self.encode(batch["input_ids"], batch["attention_mask"])
> > > > > > >         logits = self.head(pooled)
> > > > > > >         labels = batch["labels"]
> > > > > > >         loss = F.binary_cross_entropy_with_logits(logits, labels)
> > > > > > >         return {"loss": loss, "logits": logits}
> > > > > > > ```
> > > > > > >
> > > > > > > ``` bash
> > > > > > >   python scripts/train_bt_cross_encoder.py \
> > > > > > >   --model_name_or_path Qwen2.5-3B \
> > > > > > >   --data_path data.jsonl \
> > > > > > >   --output_dir Qwen2.5-3B-bt-cross \
> > > > > > >   --per_device_train_batch_size 16 \
> > > > > > >   --learning_rate 5e-6 \
> > > > > > >   --num_train_epochs 2 \
> > > > > > >   --gradient_accumulation_steps 2 \
> > > > > > >   --bf16 True \
> > > > > > >   --logging_steps 10 \
> > > > > > >   --save_steps 1000
> > > > > > > ```
> > > > > > > | **Method** | **Agreement** | **Precision** | **Recall** | **F1** |
> > > > > > > | --- | --- | --- | --- | --- |
> > > > > > > | **Cross-BT-3B (Pairwise)** | 75.18 | 50.06 | 55.97 | 52.84 |
> > > > > > > | **JudgeLRM-3B (Ours)** | **77.68** | **74.26** | **70.86** | **72.12** |
> > > > > > >
> > > > > > > The results reveal a critical insight into why reasoning is necessary:
> > > > > > >
> > > > > > > The Cross-BT model achieves relatively high agreement (75.18%), likely by overfitting to surface-level heuristics inherent in the GPT-4 training data (e.g., "length bias" or "format bias"). It mimics the teacher's superficial preferences. However, its low F1 (52.84) and near-random Precision (50.06) on the human-annotated PandaLM (OOD) indicate a failure to judge content quality. Without the explicit reasoning chain ($\langle think \rangle$), the scalar reward model lacks the "test-time compute" needed to perform verification or detailed comparison, effectively reducing it to a heuristic guesser.
> > > > > > >
> > > > > > > This negative result reinforces our core claim: abstracting judgment into a single scalar prediction is insufficient. Generative reasoning (JudgeLRM) is essential for robust, generalized evaluation.

---

### Official Review · Reviewer_miWA · 2025-10-30

**Soundness:** 1
**Presentation:** 2
**Contribution:** 2
**Rating:** 2
**Confidence:** 3

**Summary:**

This paper establishes a relationship between reasoning ability (enhanced through reinforcement learning) and judge quality. The authors first find a negative correlation between SFT performance and judge quality on reasoning-heavy tasks. They then propose RL-based rewards and train the judge model using RLVR. Experimental results show that JudgeLRMs yield significant improvements.

**Strengths:**

1. The paper introduces a new dimension of judge quality, i.e., its relationship with reasoning ability.
2. The paper adapts RLVR to pairwise response comparisons and introduces three types of content rewards.
3. The paper conducts experiments using models of various sizes and from different families to validate the effectiveness of the RL method.

**Weaknesses:**

1. The authors regard SFT as the opposite of reasoning, which is not convincing. This makes the overall claim somewhat confusing.
    - Q1: SFT models distilled from GPT-4 should also be capable of generating CoT. As shown in Figure 18, the response lengths of JudgeLRM-3B and JudgeLRM-7B are not particularly long. What, then, is the major difference between the responses produced by the SFT and RL models?
    - Q2:  What if trajectories from LRMs such as DeepSeek-R1 or the Qwen3 series are used to fine-tune the model via SFT?
2. The ablation study is not comprehensive, so the necessity of the proposed rewards cannot be fully verified. The authors introduce three types of content rewards, but the ablation results only report *w/o r_absolution + r_confidence*.
    - Q3: Could you provide more ablation studies to demonstrate that all three rewards are indispensable? I am interested in whether the policy model can learn such complex relationships from a sparse scalar reward.
3. The writing quality is relatively low. In Table 2, there is a nonsensical phrase "yaobuyao," and the table also mixes up *Instruct* and *Ins*. Lines 315–323 lack indices (4) and (5). Additionally, "Qwen3 Base" in Table 3 does not clarify whether it refers to the Qwen3-Base series or a model based on Qwen3 (the reasoning version).

**Questions:**

Q4: What are the results of Qwen3-4B and Qwen3-8B without any training? Since they are reasoning models, are they better than their non-reasoning counterparts?

Q5: Is JudgeLRM-8B based on Qwen3-8B-Base or Qwen3-8B? If it is based on Qwen3-8B-Base, why not use Qwen3-8B, given that the focus is on LRMs? If it is based on Qwen3-8B, could you explain why it performs worse than JudgeLRM-7B on PandaLM, considering that Qwen3-8B has been shown to outperform Qwen2.5 in reasoning ability?

---

> ### Author Response · Authors · 2025-11-21
>
> 1. We clarify that we do not view SFT as the "opposite" of reasoning. Rather, we argue that SFT and RL represent fundamentally different training paradigms that affect model parameter distributions distinctively. Recent work [1] suggests that RL and SFT operate on different parameter regions with distinct geometric properties. SFT focuses on **distribution matching** (mimicking the teacher's surface form), whereas RL performs **active search** within the distribution to optimize for correctness and logic. It implecates that while SFT can learn to mimic the *style* of reasoning CoT, RL is required to verify and reinforce the *validity* of the reasoning steps, which is crucial for high-fidelity judgment.
>
> Q1: What is the main difference between SFT and RL outputs?
>
> The difference is not necessarily in length, but in the emergence of specific Micro-Level Cognitive Behaviors (as detailed in our new Appendix Appendix Evaluation of Reasoning Capabilites in JudgeLRM).
>
> - **SFT (Imitation):** As shown in Table 6 (Case ID 549), the SFT model's critique is generic and repetitive (*"Both responses were relevant... response was repetitive"*). It successfully performs Induction (e.g. summarizing patterns) but fails to dig deeper.
> - **RL (Justification):** In contrast, Table 13 shows that JudgeLRM exhibits emergent behaviors like Verification (e.g. checking facts against rules), Error Identification (e.g. pinpointing specific flaws), and Decision Justification (e.g. formulating a hypothesis on why A > B).
> RL incentivizes the model to perform Abduction, constructing the "best available explanation" for a judgment, whereas SFT models often collapse into safe, generic summaries despite having similar response lengths.
>
> Q2: What about SFT on LRM traces (e.g., DeepSeek-R1)?
>
> We explicitly evaluated this scenario by incorporating both an existing open-weights model and a newly trained baseline to ensure a rigorous comparison:
>
> We compared against the official DeepSeek-R1-Distill-Qwen-7B. As shown in Table below, it achieves an F1 of 59.16 on PandaLM, which is significantly outperformed by JudgeLRM-7B (F1 75.05).
>
> To control for training variables, we trained a new baseline, SFT-Distill-R1-Think, by fine-tuning Qwen2.5-3B-Instruct on DeepSeek-R1's reasoning traces. While distilling strong reasoning traces improves performance (F1 67.33) compared to standard baselines, it still falls short of JudgeLRM-3B (F1 72.12) by a significant margin.
>
> This comparison confirms that simply fine-tuning on high-quality reasoning traces leads to "surface-level imitation" (learning the CoT format) but fails to fully master the **self-correction and verification mechanisms** that RL instills. RL is essential for the model to not just mimic reasoning, but to actively optimize the reasoning path for correct judgments.
>
> Moreover, the superior generalization of JudgeLRM (Figure 3) further proves that RL enables the model to handle harder problems that rote memorization cannot address.
>
> | Metric | Agreement | Precision | Recall | F1 |
> | --- | --- | --- | --- | --- |
> | DeepseekR1-distill-Qwen7B | 62.46 | 58.06 | 64.17 | 59.16 |
> | SFT-Distill-R1-Think (Qwen2.5-3B-Instruct)  | 73.37 | 66.45 | 70.28 | 67.33 |
> | JudgeLRM-3B | 77.68 | 74.26 | 70.86 | 72.12 |
>
> [1] The Path Not Taken: RLVR Provably Learns Off the Principals. https://arxiv.org/abs/2511.08567

---

> ### Author Response · Authors · 2025-11-21
>
> 2. We appreciate the suggestion to rigorously verify the necessity of each reward component. We have completed the full ablation study in Table 4 (updated below). The results confirm that all three rewards are indispensable and operate in a strict hierarchy of needs.
>
> | **Method** | **Agreement** | **Precision** | **Recall** | **F1** |
> | --- | --- | --- | --- | --- |
> | JudgeLRM-7B | **78.28** | **74.90** | **75.74** | **75.05** |
> | w\/o. $R_{rel}$ | 52.55 | 49.95 | 50.61 | 46.43 |
> | w. $R_{rel}$, w\/o. $R_{abs} + R_{conf}$ | 75.78 | 69.09 | 73.69 | 70.36 |
> | w. $R_{rel} + R_{conf}$, w\/o. $R_{abs}$ | 75.58 | 70.64 | 66.69 | 68.16 |
> | w. $R_{rel} + R_{abs}$, w\/o. $R_{conf}$ | 77.08 | 71.36 | 75.14 | 72.36 |
>
> - **$R_{rel}$ is Fundamental:** Removing the relation reward leads to catastrophic failure (F1 46.43), confirming that learning the comparative "win/loss" relationship is the foundation of the judging task1.
>
> - **$R_{abs}$ is the Prerequisite for Confidence:** A critical finding arises when comparing the baseline (F1 70.36) with the w\/o $ R_{abs}$ variant (F1 68.16). Adding confidence rewards *without* absolute score constraints actually hurts performance (-2.2%). It’s because our design strictly activates $R_{abs}$ and $R_{conf}$ only when the relation is correct. Without the fine-grained calibration from $R_{abs}$, the confidence reward incentivizes the model to be decisive (large score gaps) based on imprecise judgments, leading to uncalibrated behavior. This proves that precision ($R_{abs}$) is a necessary precondition for effective confidence ($R_{conf}$).
>
> - **$R_{conf}$ solves the Last Mile:** Once precision is secured (w. $R_{abs}$), adding $R_{conf}$ boosts F1 from 72.36 to 75.05. It helps the model overcome indecision in hard samples where scores are close.
>
> Regarding the reviewer's curiosity about sparse scalar rewards: We show that the policy can learn complex relationships from sparse binary signals ($R_{rel}$ only), achieving a respectable F1 of 70.36. However, to surpass teacher models and handle subtle reasoning nuances, the dense supervision provided by $R_{abs}$ and the decisiveness from $R_{conf}$ are essential to reach the peak performance of 75.05.
>
> 3. We sincerely apologize for the careless writing errors and inconsistencies. We thank the reviewer for the meticulous reading, which helped us polish the manuscript. We have uploaded a revised PDF that corrects the typos.
>
> 4. We performed the requested evaluation on the raw, untrained Qwen3-4B and Qwen3-8B models to understand the impact of intrinsic reasoning capabilities. The results on PandaLM are presented below:
>
> | **Model** | **Agreement** | **Precision** | **Recall** | **F1** |
> | --- | --- | --- | --- | --- |
> | Qwen2.5-3B-Instruct | 68.50 | 50.92 | 56.13 | 51.57 |
> | Qwen3-4B | 75.88 | 70.34 | 66.53 | 67.86 |
> | Qwen3-8B | 75.98 | 74.07 | 66.26 | 68.34 |
> | JudgeLRM-4B | 77.88 | 72.33 | 73.67 | 72.87 |
> | JudgeLRM-8B | **80.78** | **78.25** | **73.78** | **75.54** |
>
> As the reviewer hypothesized, the "reasoning-oriented" Qwen3 models significantly outperform their non-reasoning counterparts (e.g., Qwen3-4B's 67.86 vs. Qwen2.5-3B's 51.57). This confirms that intrinsic reasoning capabilities provide a better starting point for judgment tasks.
>
> However, raw reasoning capability is not enough. JudgeLRM-4B and JudgeLRM-8B significantly surpass their respective Qwen3 base models by large margins. This demonstrates that our RL training does not simply rely on the base model's knowledge. Instead, it effectively specializes the model's general reasoning capabilities into evaluative reasoning (e.g., verification and criteria mapping), unlocking performance levels that the base reasoning models cannot achieve out-of-the-box.

---

> ### Author Response · Authors · 2025-11-21
>
> 5. We clarify that JudgeLRM-8B is initialized from Qwen3-8B Instruct version (as Qwen team defines). We prioritized the Instruct version over the Base version due to training stability and convergence efficiency observed in our pilot studies:
>
> - **Stability**: The pure Base model struggled significantly to learn the structured output format. The Base model failed to produce valid structures for the first 800 steps using the $R_{struct}$ reported in the paper. To induce structural adherence, we had to aggressively scale the structure reward ($R_{struct} \times 5.0$).
> - **Convergence:** The Base model required $\sim 640$ steps for the reward to converge, whereas the Instruct version converged much faster (around step 320).
> - **Performance Gap:** We validated this on the 4B scale. Training from Qwen3-4B-Base yielded a PandaLM F1 of 69.76, which still lags behind the Instruct-based JudgeLRM-4B (F1 72.87). Thus, the Instruct model provides a more effective starting point for our RL framework.
>
> 2. Performance Anomaly (8B vs. 7B)
>
> We respectfully acknowledge that the JudgeLRM-8B performance reported in the initial submission was anomalously lower than expected given its superior backbone.
>
> This was due to computational resource constraints during the submission window, which forced us to report results from an early, under-trained checkpoint before the model had fully converged. We have since completed the training process. The fully converged JudgeLRM-8B now demonstrates the expected performance gains, outperforming the 7B model. **We have updated Table 3 in the revised PDF with these corrected results.**
>
> | Model | Agreement | Precision | Recall | F1 |
> | --- | --- | --- | --- | --- |
> | Qwen3-4B (Instruct) | 75.98 | 74.07 | 66.26 | 68.34 |
> | RL w/ Qwen3-4B-Base | 76.48 | 69.51 | 70.11 | 69.76 |
> | JudgeLRM-4B (RL w/ Instruct) | 77.88 | 72.33 | 73.67 | 72.87 |

---

> > ### Comment · Reviewer_miWA · 2025-11-26
> >
> > I appreciate the authors' efforts in adding new experimnets. Without these results, I believe the main claim of the paper would not be sufficiently supported (which motivated my initial low score). However, I still have the following concerns:
> >
> > 1. The paper claims that reasoning ability is essential for judgment and that RL training enhances this ability beyond SFT. However, this link is not convincingly demonstrated. In particular, reasoning is not well defined in this paper, nor is it demonstrated how RL specifically enhances this capability for judgment tasks (The case study in Section 4.6 may also be achieved through distillation from a reasoning model). In this sense, Appendix E provides useful direction, but I would encourage the authors to provide more detailed comparisons to support the main claim, such as between Qwen3-8B vs. JudgeLRM-8B, and between SFT-Distill-R1-Think vs. JudgeLRM-3B.
> > 2. Regarding the statement "due to computational resource constraints during the submission window, which forced us to report results from an early, under-trained checkpoint...", this *truly* raises concerns about result robustness. Repeated experiments and random seed variance are needed to ensure stability and reproducibility.
> > 3. The low GPT-4 performance on PandaLM makes me concerned about the recency and reliability of JudgeLM. It may help to evaluate on additional datasets and incorporate human verification to ensure that the conclusions hold beyond PandaLM.
> >
> > Overall, the writing needs to be improved. The RL reward design itself is interesting, but more riguous evaluation is needed to substantiate its advantage. With the additional experiments, I would like to raise my score to 4.

---

> ### Author Response · Authors · 2025-12-01
>
> We appreciate the reviewer's insightful suggestion to clarify the definition of reasoning and to rigorously distinguish the contributions of RL from SFT distillation.
>
> We’ve given the explicit definition of reasoning capabilities in Appendix E on two levels:
>
> - **Macro-Level Meta-Abilities**: Based on Gandhi et al. (2025) [1] & Hu et al. (2025) [2], we categorize reasoning into Deduction (applying criteria), Induction (synthesizing observations), and Abduction (justifying preference in ambiguous cases).
>
> - **Micro-Level Cognitive Behaviors**: These include specific actions like Verification, Error Identification, and Self-Correction.
>
> Furthermore, there may be a misunderstanding: merely possessing \<think\> tags and reasoning paths (changing the form of the reasoning path) does not equate to possessing effective reasoning capabilities. In our experiments in Table 2, we have already demonstrated that SFT-Distill-R1-Think-3B (which distills R1's paths) achieves an F1 score of 67.33 , which is inferior to JudgeLRM-3B at 72.12. Additionally, in our ablation study, we illustrated that content is more important than form, as JudgeLRM's performance is significantly higher than the ablation setting that retains only the structural reward.
>
> Regarding the extraction of reasoning steps, we found that large models often mistake ordinary descriptive sentences for reasoning and extract them. To address this, we strengthened the prompt in Appendix E (refer to updated pdf) to emphasize extracting only "conflicting" reasoning; steps are counted only when the model overturns itself or proposes competitive hypotheses. As an example, in the deduction prompt we include “
>
> - The "Wait" Test: If the sentence functions as a "Stop & Turn" signal (changing the direction of thought), extract it. If it is a "Go Ahead" signal (confirming the current path), IGNORE it.
> - Substance over Form: Do not extract sentences based on keywords like "assume" or "check". Extract them only if they contain the *content* of the counter-argument or the error found.
> - Outcome Dependency: Only extract steps that actually impacted the final judgment.”
>
> Moreover,  powerful models may over-interpret descriptive sentences as reasoning. To ensure specific and conservative measurement, we employed gpt-5-nano-2025-08-07 for extraction, leveraging its precision in instruction following to avoid "over-reading" implicit logic.
>
> We performed the requested comparison between Qwen3-8B vs. JudgeLRM-8B and SFT-Distill-R1-Think vs. JudgeLRM-3B. The results reveal a fundamental difference in how these models reason (Qwen2.5-3B-Instruct refers to the base model prompted with the structured \<think\> format).
>
> | **Model (%)** | **Deduction (e.g., Verification)** | **Induction (e.g., Synthesis)** | **Abduction (e.g., Justification)** |
> | --- | --- | --- | --- |
> | Qwen2.5-3B-Instruct | 12.4 | 48.8 | 15.8 |
> | SFT-Distill-R1-Think-3B | 31.2 | 61.6 | 26.6 |
> | **JudgeLRM-3B** | 33.2 | 57.2 | 35.1 |
> |  |  |  |  |
> | Qwen3-8B | 44.4 | 58.9 | 39.1 |
> | **JudgeLRM-8B** | 61.9 | 61.7 | 46.5 |
>
> JudgeLRM-3B outperforms the distilled model significantly in Abduction (+8.5%), which is crucial for identifying why one response is better than another. This indicates that the outcome-driven RL reward forces the model to engage in genuine Trial-and-Error and Pattern Recognition to maximize the reward, rather than just imitating the reasoning path's surface form.
>
> Notably, SFT-Distill-R1-Think-3B shows high Induction (61.6%) but significantly lower Abduction (26.6%) compared to JudgeLRM. This suggests that **SFT** successfully mimics the "linear narrative" style and summarization format of the teacher model (R1) but **lacks the depth in conflict resolution and justification**.
>
> The comparison at the 8B scale confirms this trend, with JudgeLRM-8B showing a massive jump in Deduction (+17.5%), validating that RL effectively activates the model's capability to verify facts and criteria against the prompt strictness.
>
> These findings support our claim that while distillation can transfer the format of reasoning, RL is essential for internalizing the process of **judgment-specific reasoning** (verification and justification), which directly correlates with the superior performance shown in Table 2 (JudgeLRM-3B 72.12 vs. SFT-Distill 67.33).
>
> [1] Cognitive behaviors that enable self-improving reasoners, or, four habits of highly effective stars. COLM 2025.
>
> [2] Beyond 'Aha!': Toward Systematic Meta-Abilities Alignment in Large Reasoning Models. https://arxiv.org/abs/2505.10554

---

> ### Author Response · Authors · 2025-12-01
>
> We appreciate the reviewer's scrutiny regarding result robustness and the reliability of our evaluation. We have conducted additional stochastic experiments and cited independent external validations to address these concerns.
>
> 1. Robustness and Reproducibility: Verification via Random Seed Variations To dispel concerns about result stability due to "early checkpoints" or resource constraints, we performed repeated runs with varying random seeds.
>
> While **our reported results** in the paper utilize deterministic settings (top_k=1, do_sample=False, num_beams=4) to **ensure exact reproducibility**, here we introduce randomness by setting Temperature=0.7 to strictly test stability under noise. We evaluated JudgeLRM-7B/8B across three distinct seeds (42, 123, 2024). As shown in Table below, the performance remains highly stable.
>
> | **Model** | **Seed** | **Agreement** | **Precision** | **Recall** | **F1** |
> | --- | --- | --- | --- | --- | --- |
> | **JudgeLRM-8B** | 42 | 80.28 | 76.13 | 73.39 | 74.55 |
> |  | 123 | 80.18 | 76.57 | 73.54 | 74.82 |
> |  | 2024 | 80.18 | 76.37 | 73.54 | 74.75 |
> | *Statistics* | *Mean (Std)* | *80.21 (±0.05)* | *76.36 (±0.18)* | *73.49 (±0.07)* | **74.71 (±0.11)** |
> | **JudgeLRM-7B** | 42 | 79.78 | 75.25 | 72.76 | 73.83 |
> |  | 123 | 79.98 | 75.53 | 73.15 | 74.18 |
> |  | 2024 | 80.28 | 76.64 | 73.61 | 74.89 |
> | *Statistics* | *Mean (Std)* | *80.01 (±0.20)* | *75.81 (±0.60)* | *73.17 (±0.34)* | **74.30 (±0.44)** |
>
> The standard deviation for F1 is extremely low (**±0.11 for 8B** and **±0.44 for 7B**). The variations are statistically insignificant, confirming that our model's performance is robust and not an artifact of a specific seed. It is expected that sampling (Temp=0.7) yields slightly lower scores (approx. 0.5-1.0 points) compared to the deterministic Greedy/Beam search used in our main paper due to introduced noise. The consistency under this stricter condition further validates the reliability of our reported results.
>
> 2. Timeliness and Reliability: We would like to highlight that recent works continue to validate JudgeLRM as a strong baseline:
> - **PandaLM Validation:** [1] independently evaluated JudgeLRM-7B on PandaLM, reporting an accuracy of **72.37**. This is comparable to concurrent advanced models like RM-R1[2] (72.71), demonstrating that JudgeLRM remains a robust SOTA method.
> - **Cross-Dataset Consistency:** Beyond PandaLM, JudgeLRM achieved **74.45** on **RewardBench** [1], significantly outperforming standard baselines. This cross-dataset consistency strongly supports the generalizability of our method beyond the specific benchmarks used in our paper.
>
> 3. Generalization: Regarding the concern about the performance gap between JudgeLRM and GPT-4 on PandaLM, we clarify that this demonstrates successful generalization, not data overfitting. As noted in our response to Reviewer g7Nk, PandaLM is a human-annotated, out-of-distribution (OOD) benchmark, distinct from the GPT-4-labeled JudgeLM training set. We did not perform SFT on GPT-4's explanations, which often leads to mimicking teacher errors and biases. Instead, we used labels solely as reward signals. **JudgeLRM-7B (75.05) significantly outperforms its teacher GPT-4 (61.8) on this OOD set**.  This confirms that the absolute reward successfully guided the model to acquire **generalized reasoning capabilities** that remain effective in strictly human-aligned, out-of-distribution scenarios.
>
> Moreover, as demonstrated in our Ablation Study (Table 4), removing our specific Judge-wise Outcome Reward causes performance to drop significantly (from 75.05 to 68.16). This proves the gain stems from our RL method guiding the model to generalize better than the noisy supervision provided by GPT-4.
>
> [1] OpenReward: Learning to Reward Long-form Agentic Tasks via Reinforcement Learning. https://arxiv.org/abs/2510.24636.
>
> [2] RM-R1: Reward Modeling as Reasoning. https://arxiv.org/abs/2505.02387

---

### Author Response · Authors · 2025-12-03

We sincerely thank the Area Chair and for managing the review process under these challenging circumstances.
We are also deeply grateful for the active and responsible engagement from all three reviewers throughout the rebuttal period. Your detailed feedback has significantly improved the quality and rigor of our work.

We are encouraged by the positive consensus regarding the significance of our work. We are particularly thrilled by Reviewer Nsgu’s strong endorsement (Score: 8), who highlighted that the paper possesses **"strong results and deepened analysis"** and is **"of sufficient quality for acceptance."**

We appreciate Reviewer miWA's decision to **raise the score** following our additional experiments. Regarding Reviewer g7Nk, we believe our latest response (including theoretical proofs and new baselines) has comprehensively addressed the remaining concerns about reward design and baselines.

The improvements include:

1.  **Theoretical Grounding of Reward Design:** Addressing the concerns about "heuristic" rewards, we provided a proof demonstrating that our discrete, bounded reward functions as a "Satisficing Objective". We clarified how the hierarchical dependency strictly prevents the forceful separation of ties, ensuring the stability required for GRPO training.

2.  **Clarifying SFT vs. RL with New Baselines:** To demonstrate that RL is not merely imitating reasoning traces, we added multiple baselines (including SFT-Distill-R1-Think and Pairwise Cross-BT). These experiments confirm that JudgeLRM significantly outperforms models fine-tuned on high-quality reasoning traces, proving that RL is essential for internalizing verification and self-correction mechanisms.

3.  **Robustness and Definitions:** We verified our results across multiple random seeds to ensure stability and explicitly defined reasoning capabilities (Macro-Abilities vs. Micro-Behaviors) in Appendix E.

We have incorporated all necessary changes into the revised version and expanded the Appendix. We believe the rebuttal and the revised version have adequately resolved the concerns regarding the theoretical validity of the reward function, the necessity of RL over SFT distillation, and the robustness of our results.

Thanks again for your time and dedication.

---

### Meta-Review · Area_Chair_Fj4K · 2026-01-06

**Summary:**

The paper proposes JudgeLRM, a framework for training Large Language Models (LLMs) to act as judges using Reinforcement Learning (RL). The authors introduce a specialized reward function consisting of structural, relational, absolute, and confidence components. The method is evaluated on benchmarks like PandaLM, showing performance gains over SFT baselines and larger models like GPT-4.

**Reviewer Concerns:**

Unprincipled Reward Design: Reviewer g7Nk (Score: 2) strongly criticized the reward function as "extremely heuristic," specifically noting that the confidence reward is not a proper scoring rule. While the authors provided a late rebuttal argument characterizing their reward as a "satisficing objective" (analogous to hinge loss) to ensure stability, this defense justifies the engineering utility of the heuristic rather than establishing it as a theoretically principled scoring rule. The reliance on specific thresholds (e.g., 120 tokens) and discrete bins reinforces the view that the method relies on ad-hoc tuning.

Definition of Reasoning: Reviewer miWA (Score: 4) questioned whether the model genuinely learns "reasoning" or merely imitates verbose, structured outputs. While the authors added useful distillation baselines to argue for the necessity of RL, the distinction remains subtle, and reliance on GPT-4/5 for validation introduces potential circularity.

Consensus: Contrary to the authors' claim of a "positive consensus," the reviewer scores (2, 4, 8) indicate a significant split. The negative reviews focus on fundamental methodological soundness, while the positive review focuses on empirical results.

**Reviewer Scores:**

Reviewer g7Nk (Score: 2): Did not respond to the final "satisficing objective" argument, but  g7Nk's prior comments indicate a strong demand for a principled framework (proper scoring rules) which the authors' engineering-focused defense does not provide.

Reviewer miWA (Score: 4): Acknowledged improvements but remained borderline, unconvinced about the robustness and the "reasoning" definition.

Reviewer Nsgu (Score: 8): Strongly positive based on empirical results.

---

### Decision · Program_Chairs · 2026-01-26

Reject